# Muscle abnormalities worsen after post-exertional malaise in long COVID

Brent Appelman [1,2,15], Braeden T. Charlton [3,4,15], Richie P. Goulding [3,4], Tom J. Kerkhoff [3,4,5,6], Ellen A. Breedveld [3,4], Wendy Noort [3,4], Carla Offringa [3,4], Frank W. Bloemers [4,7], Michel van Weeghel [8], Bauke V. Schomakers [8], Pedro Coelho [9,10,11], Jelle J. Posthuma [7,12], Eleonora Aronica [11], W. Joost Wiersinga [1,2,13], Michèle van Vugt [2,14,15] ✉ & Rob C. I. Wüst [3,4,15] ✉

A subgroup of patients infected with SARS-CoV-2 remain symptomatic over three months after infection. A distinctive symptom of patients with long COVID is post-exertional malaise, which is associated with a worsening of fatigue- and pain-related symptoms after acute mental or physical exercise, but its underlying pathophysiology is unclear. With this longitudinal case-control study (NCT05225688), we provide new insights into the pathophysiology of post-exertional malaise in patients with long COVID. We show that skeletal muscle structure is associated with a lower exercise capacity in patients, and local and systemic metabolic disturbances, severe exercise-induced myopathy and tissue infiltration of amyloid-containing deposits in skeletal muscles of patients with long COVID worsen after induction of post-exertional malaise. This study highlights novel pathways that help to understand the pathophysiology of post-exertional malaise in patients suffering from long COVID and other post-infectious diseases.

Chronic sequelae after acute infections contribute to debilitating conditions that affect millions worldwide[1-5]. After an acute SARS-CoV-2 infection, a subgroup of patients suffers from post-acute sequelae of COVID-19 (PASC), also called long COVID[1,6-10]. The most reported symptoms of long COVID include limited exercise tolerance and post-

exertional malaise, representing the worsening of symptoms after mental or physical exertion[1,8,10]. Current, yet unproven hypotheses explaining exercise tolerance and post-exertional malaise in long COVID include mitochondrial dysfunction, amyloid-containing deposit accumulation in blood vessels causing local hypoxia,

[1]Amsterdam UMC location University of Amsterdam, Center for Experimental and Molecular Medicine, Meibergdreef 9, Amsterdam, the Netherlands. [2]Amsterdam Institute for Infection and Immunity, Infectious diseases, Amsterdam, the Netherlands. [3]Department of Human Movement Sciences, Faculty of Behavioral and Movement Sciences, Vrije Universiteit Amsterdam, Amsterdam, the Netherlands. [4]Amsterdam Movement Sciences, Amsterdam, the Netherlands. [5]Department of Physiology, Amsterdam UMC location Vrije Universiteit Amsterdam, De Boelelaan 1117, Amsterdam, the Netherlands. [6]Amsterdam Cardiovascular Sciences, Amsterdam, the Netherlands. [7]Department of Trauma Surgery, Amsterdam UMC location University of Amsterdam, Meibergdreef 9, Amsterdam, the Netherlands. [8]Laboratory Genetic Metabolic Diseases, Core Facility Metabolomics, Amsterdam UMC location University of Amsterdam, Meibergdreef 9, Amsterdam, the Netherlands. [9]Serviço de Neurologia, Departamento de Neurociências e Saúde Mental, Hospital de Santa Maria, CHULN, Lisbon, Portugal. [10]Faculdade de Medicina, Centro de Estudos Egas Moniz, University of Lisbon, Lisbon, Portugal. [11]Department of (Neuro)pathology, Amsterdam Neuroscience, Amsterdam UMC location University of Amsterdam, Meibergdreef 9, Amsterdam, the Netherlands. [12]Flevoziekenhuis, Division of Surgery, Hospitaalweg 1, Almere, the Netherlands. [13]Division of Infectious Diseases, Department of Internal Medicine, Amsterdam UMC location University of Amsterdam, Meibergdreef 9, Amsterdam, the Netherlands. [14]Division of Infectious Diseases, Tropical Medicine, Department of Medicine, Amsterdam UMC location University of Amsterdam, Meibergdreef 9, Amsterdam, the Netherlands. [15]These authors contributed equally: Brent Appelman, Braeden T. Charlton, Michèle van Vugt, Rob C. I. Wüst. ✉e-mail: m.vanvugt@amsterdamumc.nl; r.wust@vu.nl

systemic and local inflammation, disturbed immunological responses, hormonal imbalance, and viral persistence[11–18]. The extent to which the underlying physiology of impaired exercise capacity can be separated from factors related to the onset of post-exertional malaise remains unclear, largely due to indirect assessment of the underlying biomedical and psychological parameters, the cross-sectional nature of most studies, and patient heterogeneity.

In this study, we systematically induced post-exertional malaise in a cohort of 25 well-defined patients with long COVID and controls. We obtained blood and skeletal muscle biopsies before and after a maximal exercise test (Supplemental Fig. 1) with the aim to study the biological factors contributing to the limited exercise capacity and post-exertional malaise in long COVID. Results were compared with those obtained from 21 age- and sex-matched controls who fully recovered from a mild SARS-CoV-2 infection (Table 1). We characterized Long COVID based on the criteria established by the World Health Organization, and an important inclusion criterion was the presence of post-exertional malaise[19]. Both groups were healthy and socially active prior to the initial PCR-proven SARS-CoV-2 infection. No participants were hospitalized due to SARS-CoV-2 infection. Fatigue questionnaires and accelerometer data confirmed the impact of long COVID on daily life (Supplemental Fig. 2). We show that skeletal muscle structure is associated with a lower exercise capacity in patients and that local and systemic metabolic disturbances, severe exercise-induced myopathy and tissue infiltration of amyloid-containing deposits in skeletal muscles of patients with long COVID worsen after induction of post-exertional malaise.

## Results

### Limited exercise capacity in long COVID

All participants performed a cardiopulmonary exercise test on a cycle ergometer. Maximal oxygen uptake ($\dot{V}O_{2max}$) and peak power output were substantially lower in long COVID patients (Fig. 1A, B), despite marked inter-patient heterogeneity. Patients with long COVID had lower maximal ventilation and lower maximal end-tidal partial pressure of $CO_2$ (PETCO$_2$) implying poorer ventilatory function during exercise (Supplemental Fig. 3A–C)[20–22]. The cardiovascular system was not compromised in long COVID patients (Supplemental Fig. 3D, E), suggesting that this system does not explain the limited exercise capacity in patients with long COVID. The lower maximal O$_2$ pulse (i.e. the product of stroke volume and arteriovenous O$_2$ difference; Supplemental Fig. 3F), gas exchange threshold (Fig. 1C), and peripheral O$_2$ extraction (determined via near-infrared spectroscopy; Fig.1D, E, Supplemental Fig. 3I) during exercise all indicated peripheral skeletal

muscle impairments in patients. The lower $\dot{V}O_{2max}$ in patients was not due to submaximal effort during the exercise test, as the proportion of participants attaining a plateau in $\dot{V}O_2$ did not differ between groups, the gas exchange threshold was at similar percentages of $\dot{V}O_{2max}$, and secondary criteria ([lactate], maximal RER and heart rate) were attained in both groups (Supplemental Fig. 3E, G, H).

Next, we assessed skeletal muscle structure and function to explain the lower exercise capacity in patients. Capillary density and the capillary-to-fiber ratio were not different between groups (Fig. 2A, B). However, patients exhibited a trend towards lower capillary-to-fiber ratios ($p = 0.08$, degrees of freedom 46), and capillary-to-fiber ratio correlated to $\dot{V}O_{2max}$ in both groups with identical slope and intercept (Fig. 2C). Compared to healthy controls, we observed a higher proportion of highly fatigable glycolytic fibers in the vastus lateralis muscle in long COVID patients (Fig. 2D, Supplemental Fig. 4A–G), along with a lower cross-sectional area of fatigue-resistant type I fibers in females only (Supplemental Fig. 4D–F). Fiber cross-sectional area was positively associated with maximal power output in both groups (Fig. 2E), albeit with a significantly lower intercept for patients. For a given skeletal muscle size, patients did not attain the same peak power output, suggesting that exercise performance in patients was at least partly explained by intrinsic alterations in skeletal muscle force and fatigue characteristics. Succinate dehydrogenase (SDH) activity (a marker for mitochondrial content) correlated with $\dot{V}O_{2max}$ in healthy controls, but not in patients (Fig. 2F). While patients had a significantly lower oxidative phosphorylation capacity (Fig. 3A, ED 5A–H), no differences were observed in SDH activity (Fig. 3B, C; Supplemental Fig. 5I, J), suggestive of qualitatively lower mitochondrial respiration rather than a lower mitochondrial enzyme activity. Collectively, our data indicate that the lower exercise capacity in long COVID patients is associated with a greater proportion of high-fatigable glycolytic fibers and lower mitochondrial function, with a possible additional limitation of a lower capillarization and the ventilatory system.

### Metabolic dysfunction and post-exertional malaise

To understand peripheral factors contributing to the development of post-exertional malaise, we obtained vastus lateralis muscle biopsies before and one day after the induction of post-exertional malaise. All long COVID patients experienced post-exertional malaise following maximal exercise, despite considerable heterogeneity in exercise capacity. Symptoms included muscle pain, greater severity of fatigue, and cognitive symptoms up to 7 days after maximal exercise (Table 2, Supplemental Fig. 2A, B). To systematically assess whether metabolic and mitochondrial dysfunction is associated with the pathophysiology

## Table 1 | Participant characteristics

| | Healthy controls | Long COVID | *p*-value |
|---|---|---|---|
| Number of participants | 21 | 25 | |
| *Demographics* | | | |
| Age, mean (SD), years | 42 (10) | 41 (11) | 0.82 |
| Sex, male (%) | 10 (47.6) | 12 (48.0) | >0.99 |
| BMI, mean (SD), kg/m² | 24.14 (3.37) | 25.60 (4.62) | 0.24 |
| Charlson Comorbidity Index, median [IQR] | 0 [0, 0] | 0 [0, 1] | 0.76 |
| Hospitalization for COVID-19 | 0 (0) | (0) | >0.99 |
| SARS-CoV-2 vaccination before participation (%) | 21 (100) | 24 (96.0) | >0.99 |
| Weekly working hours before initial SARS-CoV-2 infection, median [IQR] | 36 [32, 40] | 36 [32, 45] | 0.520 |
| Weekly working hours during the study, median [IQR] | 36 [32, 40] | 5 [0, 12] | <0.001 |
| Time since initial SARS-CoV-2 infection, median [IQR], days | 142 [105, 405] | 545 [455, 686] | 0.001 |
| Time since latest SARS-CoV-2 infection, median [IQR], days | 135 [93, 197] | 504 [329, 665] | 0.002 |

Continuous parametric data were analyzed using a two-sided *t*-test and are shown as mean with SD. Continuous non-parametric data were analyzed using a two-sided Wilcoxon-test and are shown as a median with an interquartile range. *p*-values were not corrected for multiple testing. Source data are provided in the Source Data File.

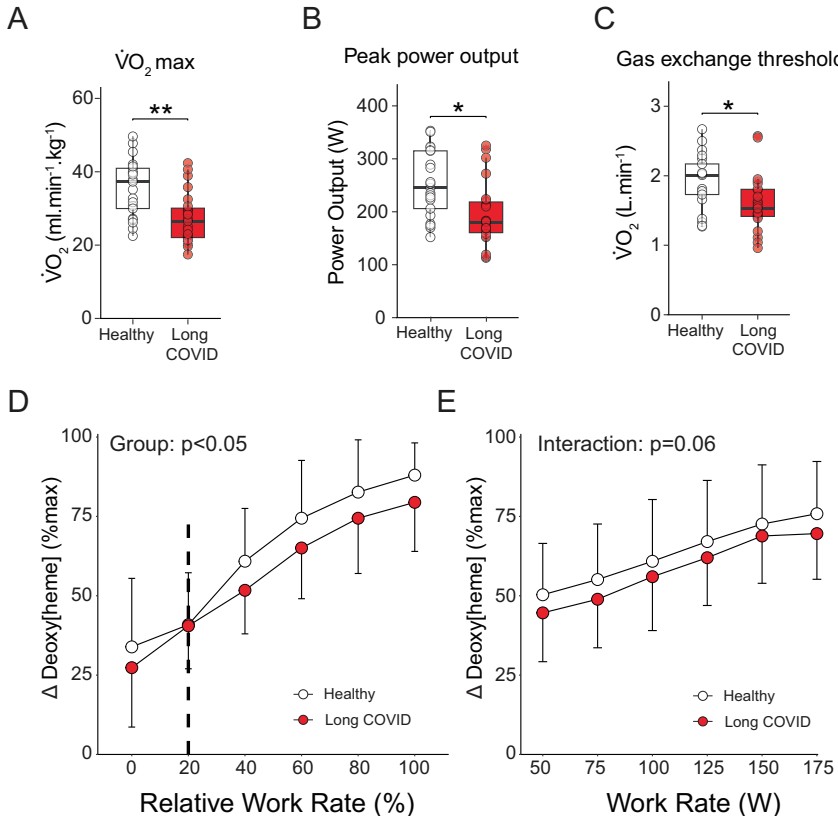

**Fig. 1 | Lower exercise capacity in patients with long COVID.** Maximal pulmonary oxygen uptake ($\dot{V}O_{2max}$, **A**, $n = 23$ long COVID, $n = 21$ healthy control), peak power output (**B**, $n = 25$ long COVID, $n = 21$ healthy control) and gas exchange threshold (**C**, $n = 23$ long COVID, $n = 21$ healthy control) were all lower ($p < 0.0001$, $p = 0.001$ and $p = 0.014$, respectively) in patients with long COVID compared to healthy controls. **D** and **E** Muscle deoxygenated [heme] responses (mean ± SD) measured by near-infrared spectroscopy were lower ($p = 0.023$) in long COVID ($n = 16$), indicative of lower peripheral oxygen extraction during exercise compared to healthy controls ($n = 18$; excessive adipose tissue precluded data analysis in remaining participants). Continuous parametric data were analyzed using a two-sided $t$-test (panels **A**–**C**). Continuous parametric longitudinal data (mean ± SD; panels **D** and **E**) were analyzed with a generalized linear mixed model. $p$-values for panel **D**, **E** were determined with a two-sided ANOVA test. Dashed line (**D**) represents the average starting point of the exercise test. *$p < 0.05$; **$p < 0.001$. Box plots show the median (centerline), the first and third quartiles (the lower and upper bound of the box), and the whiskers show the 1.5× interquartile range. Source data are provided in the Source Data File.

of post-exertional malaise[23–26], we measured mitochondrial respiration and metabolomic signatures in skeletal muscle before and one day after the induction of post-exertional malaise. Mitochondrial respiration was assessed in hyperoxic conditions to avoid diffusion limitation for oxygen that could contribute to exercise intolerance in vivo. Oxidative phosphorylation capacity decreased one day following the maximal exercise in both controls and patients (Fig. 3A). SDH activity was not reduced in healthy controls one day after exercise, but was reduced in the long COVID patients, suggesting that the combination of a reduced maximal mitochondrial respiration and decreased mitochondrial content are part of the pathophysiology of post-exertional malaise.

To better understand the resting skeletal muscle metabolism during post-exertional malaise, we annotated 116 metabolites in skeletal muscle and 83 metabolites in venous blood (Fig. 3D, E, Supplemental Fig. 6A, B). Skeletal muscle biopsies were obtained at rest, and while skeletal muscle glycolytic metabolites displayed few group differences, key metabolites of the tricarboxylic acid (TCA) cycle (including glutamate, $FAD^+$, alpha-ketoglutarate and citric acid) were lower in skeletal muscle of long COVID patients (Fig. 3D). The ratio of citric acid to lactate in skeletal muscle was lower in long COVID patients, indicative of a shift away from oxidative metabolism in patients (Supplemental Fig. 6C, D)[27]. Skeletal muscle creatine concentrations were lower in patients with long COVID (Supplemental Fig. 6A), likely contributing to the lower oxidative phosphorylation

capacity in patients. We also observed lower S-adenosylmethionine (SAM) in patients, possibly linking to reduced methylation and SAM cycle activity in long COVID patients. Dihydroxyacetone phosphate, an important intermediate in lipid biosynthesis and glycolysis was reduced in patient skeletal muscle following post-exertional malaise (Fig. 3D). The lower hydroxyphenyl acetic acid in patients is typically associated with increased mitochondrial production of reactive oxygen species (Supplemental Fig. 6A, B)[28]. Metabolites related to pyrimidine and purine synthesis (such as ATP), which is a metabolically demanding process[29], tended to be lower between healthy controls and long COVID patients upon the induction of post-exertional malaise (Fig. 3D, Supplemental Fig. 6A). Many amino acids were not different between groups at rest, but tended to be lower in patients upon the induction of post-exertional malaise (Supplemental Fig. 6A). Glycolytic metabolites in the venous blood were significantly higher, but pyruvate and other TCA cycle metabolites were lower in patients compared to controls (Fig. 3E). The induction of post-exertional malaise led to a reduction in blood glycolytic metabolites after one week, without changes in TCA cycle metabolites in long COVID patients. Various blood metabolites within the purine pathway were lower in patients compared to controls (Supplemental Fig. 6B) but did not change with the induction of post-exertional malaise. From these analyses, we conclude that TCA cycle metabolites were lower in skeletal muscle and blood in long COVID patients, but did not change during post-exertional malaise. Venous blood glycolytic metabolites were higher at

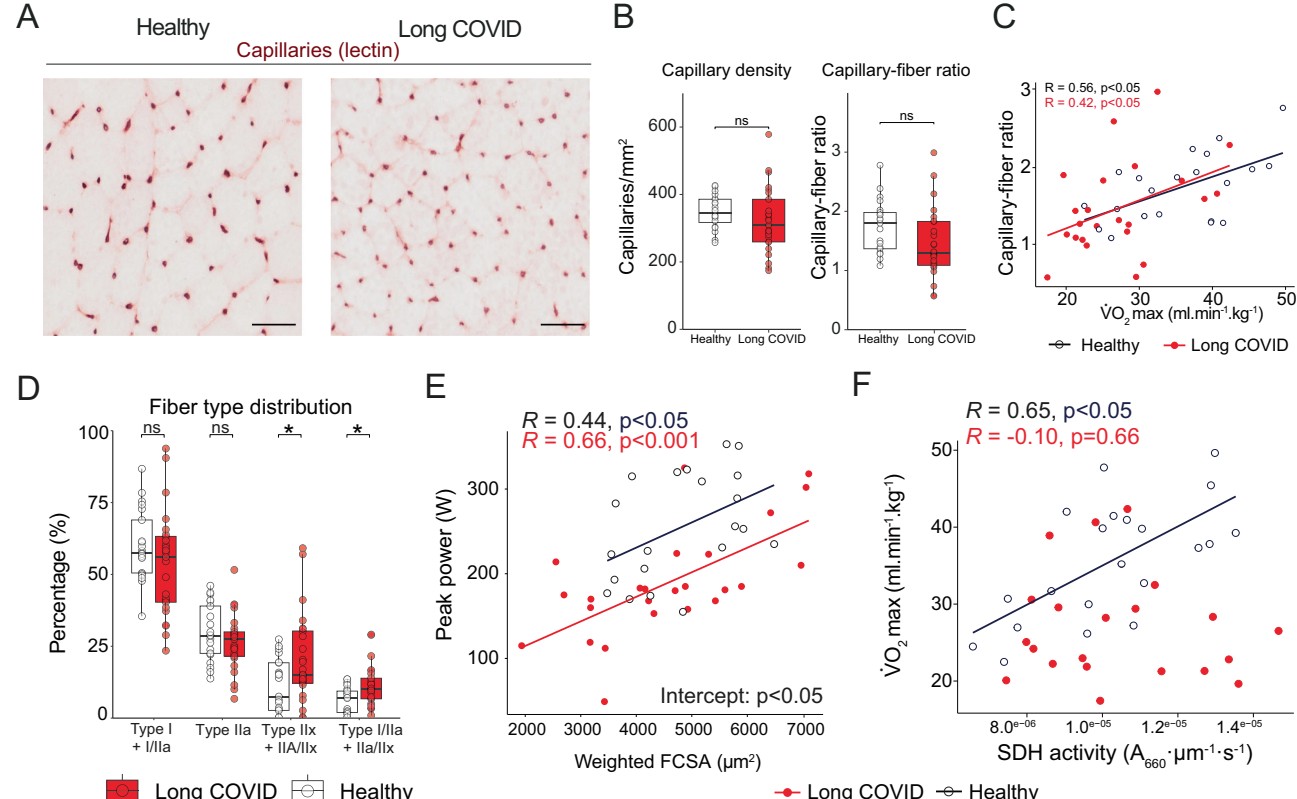

**Fig. 2 | Skeletal muscle alterations are associated with exercise capacity in patients with long COVID. A** and **B** examples of skeletal muscle capillaries; no group-differences in capillary density ($p = 0.11$) or capillary:fiber ratio ($p = 0.08$) were observed ($n = 26$ long COVID, $n = 21$ healthy control). **C** A significant association was found between capillary-to-fiber ratio and $\dot{V}O_{2max}$ for both groups ($n = 23$ long COVID, $p$-value: 0.048, $n = 21$ healthy control, $p$-value: 0.007). **D** Patients with long COVID ($n = 25$) had a higher percentage ($p$-value: 0.036) of glycolytic type IIx compared to healthy controls ($n = 21$). **E** For a given fiber cross-sectional area (FCSA), patients with long COVID ($n = 25$) had a significantly lower peak power output ($p$-value: 0.045) as compared to healthy individuals ($n = 21$). **F** Succinate dehydrogenase (SDH) activity in sections (see also Fig. 3) was associated with

maximal oxygen uptake consumption ($\dot{V}O_{2max}$) in healthy controls ($n = 21$, $p$-value: 0.0014), but not in long COVID patients ($n = 23$, $p$-value: 0.66), with significant different correlation coefficients. Continuous parametric data were analyzed using a two-sided t-test (**B, D**). Correlations were calculated using two-sided Pearson (**C, E, F**). The difference in intercept was calculated with a linear regression using a two-sided ANOVA (**E**). Correlation coefficients were compared using the R package *cocor*. *$p < 0.05$; **$p < 0.001$. Bar: 100 μm. Box plots show the median (centerline), the first and third quartiles (the lower and upper bound of the box), and the whiskers show the 1.5× interquartile range. Source data are provided in the Source Data File.

---

baseline and during post-exertional malaise but decreased one week after the induction of post-exertional malaise.

### Exercise-induced amyloid-containing deposit accumulation in skeletal muscle

It has been hypothesized that amyloid-containing deposits in the circulation can block local perfusion in long COVID, causing ischemia-reperfusion injury[30,31]. We studied whether amyloid-containing deposits were present in the skeletal muscle of long COVID patients and whether the indication of post-exertional malaise changed the concentration. We demonstrate that the concentration of amyloid-containing deposits was greater in the skeletal muscle of long COVID patients at baseline, and increased similarly in both groups after the induction of post-exertional malaise (Fig. 4A, B). As an additional control group, skeletal muscle biopsies obtained before the outbreak of SARS-CoV-2 had similar amounts of amyloid-containing deposits as our healthy control group, confirming a basal, non-SARS-CoV-2-specific abundance of amyloid-containing deposits in healthy controls[13]. We next studied the location of these amyloid-containing deposits. Visualizing amyloid-containing deposits together with capillaries or lymph vessels revealed that the skeletal amyloid-containing deposits were not located in capillaries or lymphatic vessels, but rather next to capillaries and in the extracellular matrix between muscle fibers (Fig. 4A, C, D). Amyloid-containing deposits did not overlap with cell nuclei

suggesting that these deposits are located outside infiltrating (immune) cells. We conclude that amyloid-containing deposits are not present within capillaries. Neither did we observe any signs of skeletal muscle tissue hypoxia, as the skeletal muscle capillary-to-fiber ratio, capillary density (Fig. 2B), and intracellular and circulating lactate concentrations (Supplemental Figs. 3G, 6A) were not different between long COVID patients and controls. Therefore, we conclude that post-exertional malaise cannot be explained by the hypothesis that these deposits block vessel perfusion, causing local tissue hypoxia[30,32]. The underlying reason for the increased intramuscular accumulation of amyloid-containing deposits during post-exertional malaise remains elusive.

### Exercise-induced myopathy in long COVID

To further elucidate the pathophysiology of increased muscle weakness, fatigue, and pain after exercise in long COVID patients, we determined whether specific pathological features were present in skeletal muscle before and after the induction of post-exertional malaise. As the biopsy size was variable between participants and timepoints, we scored biopsies negative/positive for pathological and immunological parameters. A larger percentage of long COVID patients displayed small atrophic fibers and focal necrosis (Fig. 5A, B), which increased significantly after exercise, indicating an exacerbated tissue damage response in patients with long COVID. Since skeletal

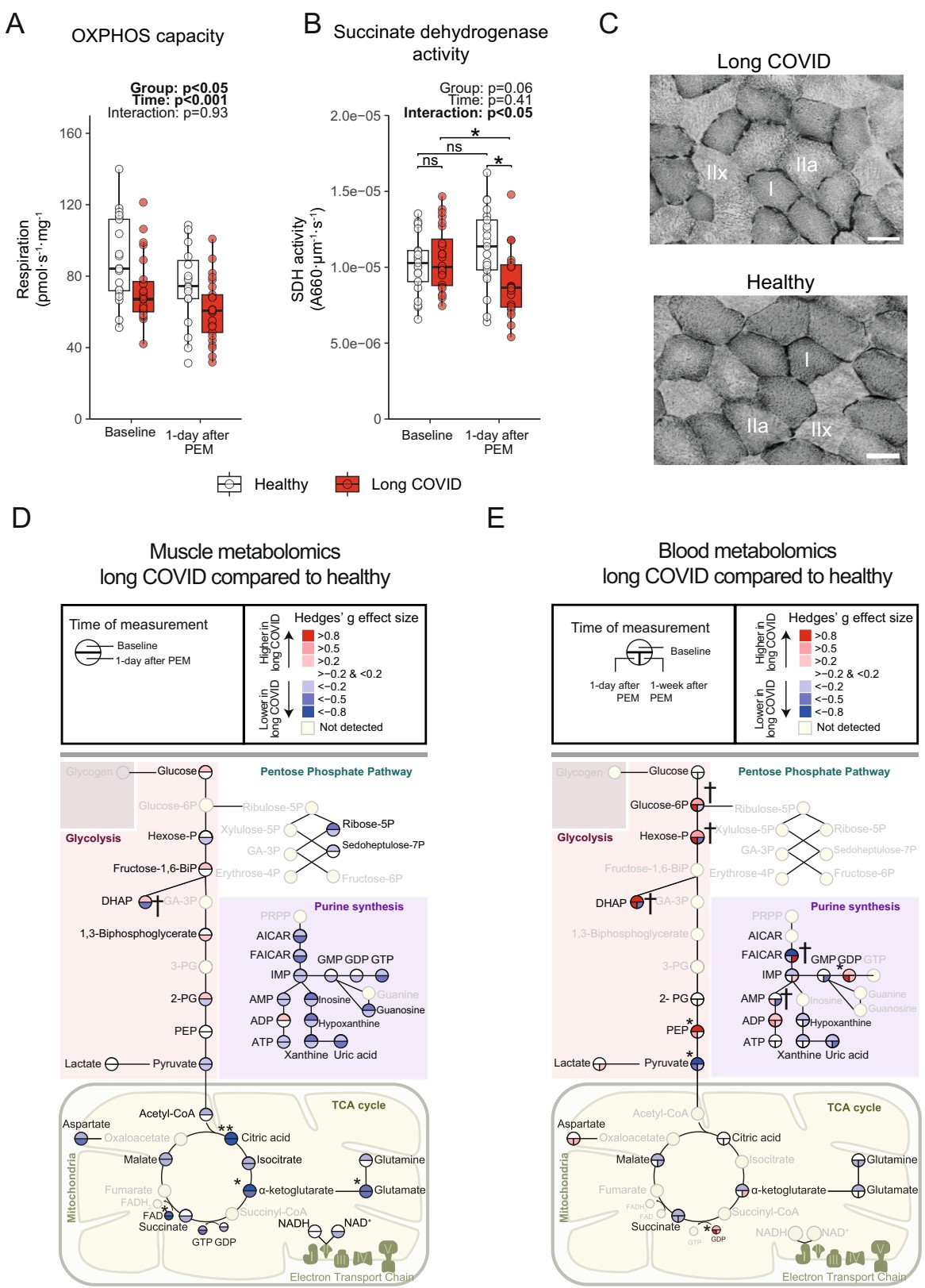

muscle is a plastic tissue, signs of skeletal muscle regeneration, such as centralized nuclei (Fig. 5C), were more evident in long COVID patients, also before the induction of post-exertional malaise. Acutely regenerating fibers, evidenced by central nuclei and a basophilic cytoplasm, were seen in biopsies from both groups (Fig. 5D), and exhaustive exercise increased the proportion of regenerating fibers in both healthy controls and long COVID patients without group differences. We

conclude that severe exercise-induced muscle damage and subsequent regeneration are associated with the pathophysiology of post-exertional malaise, and can possibly explain muscle pain, fatigue, and weakness in patients with long COVID experiencing post-exertional malaise.

Next, we explored the infiltration of immune cells (macrophages, T- and B-cells) in skeletal muscle upon exhaustive exercise. More long

**Fig. 3 | Metabolic and mitochondrial dysfunction in long COVID patients worsens with post-exertional malaise. A** Oxidative phosphorylation (OXPHOS) capacity was significantly lower in patients with long COVID ($n = 25$) compared to healthy controls ($n = 21$), and remained lower one day after induction of post-exertional malaise (PEM) in patients (Group: $p = 0.003$, Time: $p < 0.001$). **B** Succinate dehydrogenase (SDH) activity, a marker for mitochondrial density, was not different between groups ($p = 0.06$) and only reduced ($p = 0.0083$) after induction of post-exertional malaise in long COVID patients ($n = 25$) compared to healthy controls ($n = 21$). A typical example of the SDH activity is shown in panel **C**. Skeletal muscle (**D**) and venous (**E**) metabolome pathways indicate slightly higher levels of metabolites related to glycolysis, and a lower abundance of metabolites related to purine synthesis and the tricarboxylic acid (TCA) cycle, indicative of a lower reliance on oxidative metabolism in patients with long COVID ($n = 25$, both timepoints) as compared to healthy ($n = 19$, both timepoints). Faded names were not measured and shown for clarity. A higher effect size in long COVID is shown in red, lower effect size in blue. Continuous parametric longitudinal data (panels **A**, **B**, **D**, **E**) were analyzed with a generalized linear mixed model with a two-sided ANOVA. Post-hoc tests comparing each group were performed when the interaction term was significant and was performed using *emmeans* with BH adjustment (panels **A** and **B**). Effect sizes (**D** and **E**) were calculated with Hedges' *g* *$p < 0.05$; **$p < 0.001$; †$p < 0.05$ indicates a significant interaction effect. Bar: 50 μm. PEM post-exertional malaise. Box plots show the median (centerline), the first and third quartiles (the lower and upper bound of the box), and the whiskers show the 1.5× interquartile range. Source data are provided in the Source Data File.

**Table 2 | Patient-reported symptoms before and during post-exertional malaise**

| Symptoms | Baseline | 1 day after induction of post-exertional malaise | 1 week after induction of post-exertional malaise | *p*-value |
|---|---|---|---|---|
| | $n = 25$ | $n = 25$ | $n = 25$ | |
| Fatigue | 25 (100.0) | 25 (100.0) | 24 (96.0) | 0.36 |
| Self-reported fatigue severity intensity (1–10), median [IQR] | 6.00 [4.00, 7.00] | 7.00 [7.00, 8.00] | 7.00 [6.00, 7.25] | 0.001 |
| Cognitive problems | 20 (80.0) | 21 (84.0) | 23 (92.0) | 0.47 |
| Self-reported cognitive severity intensity (1–10), median [IQR] | 4.00 [3.00, 6.25] | 6.00 [5.00, 8.00] | 6.00 [4.00, 7.00] | 0.06 |
| Muscle or joint pain | 16 (64.0) | 20 (80.0) | 15 (60.0) | 0.276 |
| Chest pain | 8 (32.0) | 11 (44.0) | 8 (32.0) | 0.59 |
| Dyspnea | 6 (24.0) | 4 (16.0) | 7 (28.0) | 0.59 |
| Anosmia | 3 (12.0) | 3 (12.0) | 3 (12.0) | >0.99 |
| Sore throat | 2 (8.0) | 5 (20.0) | 3 (12.0) | 0.45 |
| Cold-like symptoms | 1 (4.0) | 0 (0.0) | 2 (8.0) | 0.35 |
| Cough | 0 (0.0) | 1 (4.0) | 1 (4.0) | 0.60 |

Continuous parametric data were analyzed using a two-sided *t*-test and are shown as mean with SD. Continuous non-parametric data were analyzed using a two-sided Wilcoxon-test and are shown as median with interquartile range. *p*-values were not corrected for multiple testing. Source data are provided in the Source Data File.

COVID patients had CD68+ macrophage infiltration in skeletal muscle compared to healthy controls (Fig. 5E). Exercise caused a similar increase in infiltrating macrophages in both groups. CD3+ T-cells were absent in healthy controls, but present in patients with long COVID before induction of post-exertional malaise (Fig. 5F). Exercise resulted in an accumulation of CD3+ T-cells inside skeletal muscle tissue, but this response was blunted in patients experiencing post-exertional malaise. CD20+ B-cells were not different between groups and tended to be higher one day after exercise (Fig. 5G).

We reasoned that the exercise-induced damage and infiltration of immune cells might be due to mitochondrial DNA (mtDNA) fragments circulating in the systemic circulation, that act as a damage-associated molecular pattern triggering a systemic inflammatory response[33–36]. However, we did not find any evidence of an exacerbated circulating mtDNA response in plasma from patients compared to controls (Supplemental Fig. 7A–F). Neither did we observe an increase in muscle breakdown products, such as creatinine and creatine kinase, in the plasma of both groups (Supplemental Figs. 6B and 7G, H). Circulating cortisol concentrations, critical for mediating homeostatic stress responses, were not different between groups at the time of measurement, with and without adjustments for time since waking (Supplemental Fig. 7I).

**SARS-CoV-2 nucleocapsid presence in skeletal muscle**
To determine whether viral remnants play a role in the differential immune response in skeletal muscle in long COVID upon induction of post-exertional malaise[1,37], we stained all muscle sections for SARS-CoV-2 nucleocapsid protein. We reasoned that if viral persistence plays a role, then the SARS-CoV-2 nucleocapsid protein, which is specific to SARS-CoV-2 and not influenced by vaccinations, should be at least present. While we did not observe any positive staining in samples obtained before the outbreak of SARS-CoV-2 (Fig. 6A, B), we found SARS-CoV-2 nucleocapsid protein in almost all patients and healthy controls (Fig. 6A, B). Despite a greater duration since the latest infection in long COVID patients, the amount of SARS-CoV-2 nucleocapsid protein was not different between groups (Fig. 6A), before and after adjusting for time since the latest infection. Participants with an infection >6 months prior to study enrollment showed no differences in the abundance of SARS-CoV-2 nucleocapsid protein, but we cannot exclude additional unknown asymptomatic infections. The SARS-CoV-2 nucleocapsid protein was not located inside muscle fibers, but rather in the extracellular matrix, however, its presence did not consistently overlap with cell nuclei. There was no correlation between the presence of SARS-CoV-2 nucleocapsid protein and amyloid-containing deposits ($R^2 = 0.03$). The concentration of SARS-CoV-2 nucleocapsid protein was not altered after the development of post-exertional malaise. Our results suggest that the presence of residual SARS-CoV-2 nucleocapsid protein is similar between patients and healthy controls, and therefore does not explain the limited exercise capacity or development of post-exertional malaise in patients with long COVID.

## Discussion
We aimed to disentangle the pathophysiology of a lower exercise capacity and post-exertional malaise in patients with long COVID and reveal distinct pathophysiological abnormalities in skeletal muscle and blood in long COVID patients using a longitudinal, case-controlled,

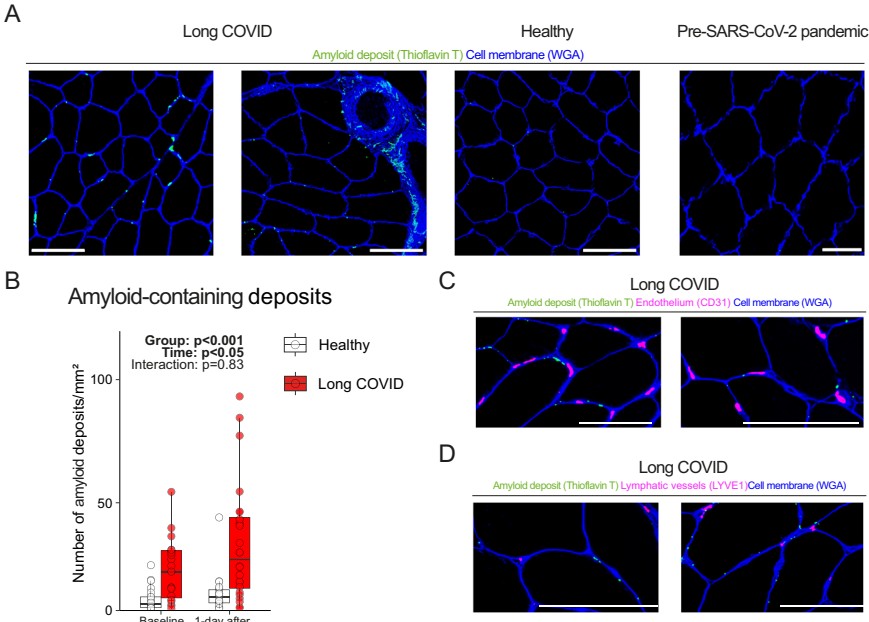

**Fig. 4 | More amyloid-containing deposits in skeletal muscle, but not located inside capillaries or lymphatic vessels. A** Typical example of amyloid-containing deposits in skeletal muscle. Pre-SARS-CoV-2 pandemic skeletal muscle sections stained with Thioflavin T showed similar levels of amyloid-containing deposits as healthy controls. **B** The concentration of amyloid-containing deposits in skeletal muscle in long COVID patients ($n = 24$) was higher (Group: $p < 0.001$) than in healthy controls ($n = 21$) and increased in long COVID patients and healthy controls upon exercise (Time: $p = 0.008$). **C** Amyloid-containing deposits were not found inside endothelial cells, but rather next to endothelial cells, or in the extracellular matrix between fibers. We performed the amyloid stainings on all patients and show a typical example in the figures. **D** Amyloid-containing deposits were not located inside lymphatic vessels. Continuous parametric longitudinal data (panel **B**) were analyzed with a generalized linear mixed model with a two-sided ANOVA. Bar: 100 µm. PEM post-exertional malaise. Box plots show the median (centerline), the first and third quartiles (the lower and upper bound of the box), and the whiskers show the 1.5× interquartile range. Source data are provided in the Source Data File.

cohort design. Patients with long COVID displayed a markedly lower exercise capacity, which related to skeletal muscle metabolic alterations and a shift towards more fast-fatigable fibers. The pathophysiology of post-exertional malaise includes an acute exercise-induced reduction in skeletal muscle mitochondrial enzyme activity, an increased accumulation of amyloid-containing deposits in skeletal muscle, signs of severe muscle tissue damage, together with a blunted exercise-induced T-cell response in skeletal muscle. Collectively, these findings help to decipher the underlying physiology of fatigue and a limited exercise capacity from the development of post-exertional malaise in patients with long COVID.

A low exercise capacity is one of the hallmark signs of long COVID, associated with a substantial burden for daily life[21]. The ventilatory and central cardiovascular system did not limit exercise capacity in long COVID patients, but our results confirm previous suggestions of a peripheral impairment in skeletal muscle metabolism in long COVID patients[21,23,38,39]. Consistent with previous hypotheses[21,38], we demonstrate that long COVID is associated with a lower skeletal muscle oxidative phosphorylation capacity. The occurrence of more IIA/X hybrid fibers and rare I/IIA hybrid fibers in patients with long COVID suggests that the pathophysiology of long COVID includes a fiber-type shift towards a less oxidative, more glycolytic phenotype. Metabolic and structural alterations are supported by venous blood and skeletal muscle metabolomes of long COVID patients, in which glycolytic metabolites were higher and TCA cycle metabolites were lower. We observed that a fiber-type shift and lower mitochondrial respiration are associated with reduced exercise capacity in patients, but these do not necessarily contribute to the pathophysiology of post-exertional malaise, as fiber-type shifts occur at a very slow pace and could have been present before long COVID. Our participants showed large variability in exercise tolerance ($\dot{V}O_{2max}$ and peak power), fiber type composition, and mitochondrial enzyme activity, but despite this large heterogeneity, all patients experienced post-exertional malaise one day after the exercise test. As such, we conclude that the pathophysiology of fatigue and a reduced exercise capacity is distinct from the rapid development of post-exertional malaise in long COVID patients. The development of post-exertional malaise could in turn, however, lead to a further reduction in exercise capacity in patients, as the acute reduction in mitochondrial SDH activity, occurrence of tissue necrosis, and possibly intramuscular accumulation of amyloid-containing deposits could worsen skeletal muscle metabolism and force production over time, causing a vicious downward circle.

Tissue hypoxia, induced by amyloid-containing deposit accumulation in the smallest capillaries has been suggested to contribute to long COVID symptoms, such as fatigue[30,40]. We found an impaired peripheral oxygen extraction (and relatively lower deoxygenation) and a higher concentration of amyloid-containing deposits in skeletal muscle, but we did not find evidence of blocked capillaries within the skeletal muscle. Our findings therefore do not support the hypothesis of chronic tissue hypoxia due to vessel blockage contributing to the development of skeletal muscle-related symptoms in long COVID or post-exertional malaise. We only evaluated the presence of amyloid, but not the function of the endothelium, or local blood flow. As such, the possible association of chronic endotheliitis or reduced blood flow to the development of post-exertional malaise and/or exercise intolerance in long COVID remains open[41,42]. Further studies into the composition of these amyloid-containing deposits and the possible role of entrapped (auto-)antibodies[43] in relation to post-exertional malaise and long COVID are warranted.

Both groups presented with more regenerating fibers following the second biopsy, indicating a possible effect of the first biopsy. Despite this, long COVID patients had more internal nuclei, atrophic

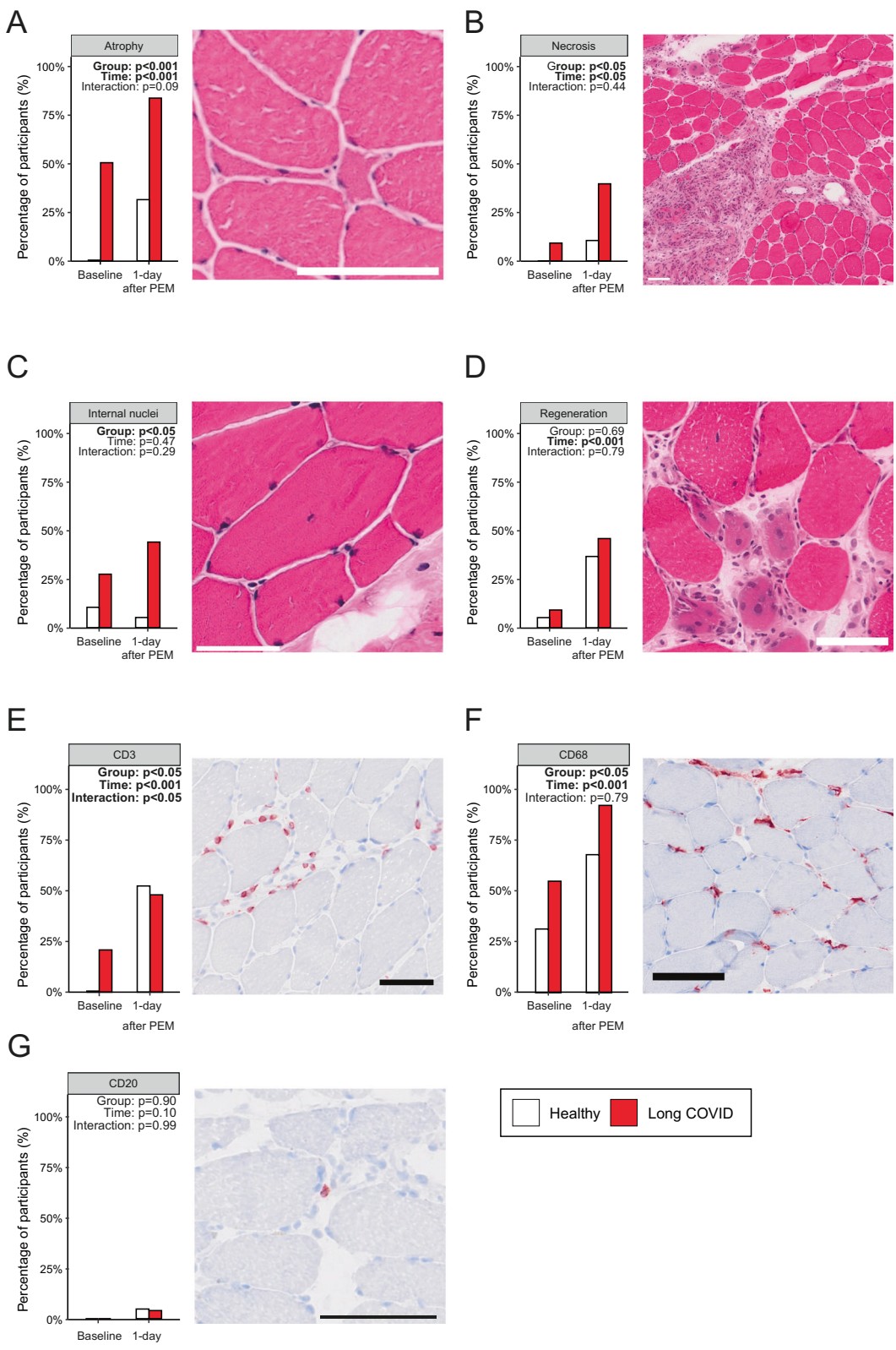

fibers, and focal necrosis after induction of post-exertional malaise compared to healthy controls. Skeletal muscle atrophy and focal necrosis can occur during severe acute SARS-CoV-2 infection[44,45], but this is the first study providing evidence of severe tissue damage upon acute exercise in long COVID patients. The higher skeletal muscle infiltration of CD68+ macrophages and CD3+ T-cells suggests a locally disturbed immune response in patients with long COVID.

Viral infections can alter mitochondrial function, and multiple studies have shown that residual SARS-CoV-2 protein presence is associated with long COVID[43,46,47]. During acute SARS-CoV-2 infection, the level of SARS-CoV-2 nucleocapsid protein rises sharply[48], and although our patients suffered from exercise-induced myopathy, there were no signs of a higher presence of SARS-CoV-2 nucleocapsid protein in patients with long COVID before or after induction of post-

**Fig. 5 | Pathological features in skeletal muscle in patients with long COVID.**
**A** Very small and angulated atrophic fibers were more abundant in patients with long COVID and in the post-exercise biopsy in both groups (Group: $p < 0.001$). **B** Large areas of necrotic fibers were observed in 36% of patients with long COVID after exhaustive exercise (Group p-value: 0.09, as compared to healthy controls). **C** Internal nuclei, indicative of fiber repair, were significantly more abundant (Group: $p = 0.002$) in the skeletal muscle of patients with long COVID, but did not increase during post-exertional malaise (PEM). **D** Regenerative fibers were not different between groups, but were more abundant in the post-exercise biopsy

(Time: $p < 0.001$). **E**: More patients with long COVID had CD3+ T-cell infiltration (Group: $p = 0.046$). **F** The presence of CD68+ macrophages was higher in long COVID patients (Group: $p = 0.03$). **G** CD20+ B-cells were not abundantly present in skeletal muscle. All panels: $n = 25$ long COVID, $n = 21$ healthy controls. Categorical longitudinal data were analyzed using logistic regression with "time" as a covariate. Post-hoc comparisons (**E**) were using the Empirical Mean Differences using R package *emmeans* with BH adjustment. Bar: 100 μm. Abbreviation: PEM; post-exertional malaise. Source data are provided in the Source Data File.

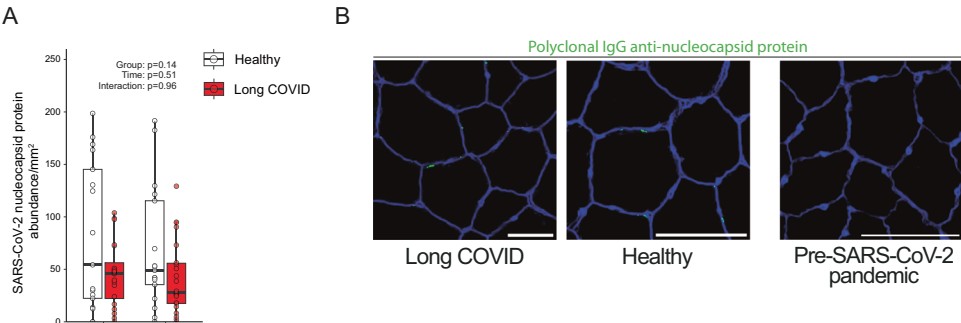

**Fig. 6 | Similar concentrations of nucleocapsid protein in skeletal muscle in patients with long COVID as control. A** The SARS-CoV-2 nucleocapsid protein was present in almost all participants, but not more abundant in long COVID ($n = 23$) compared to healthy controls ($n = 19$). **B** Typical examples of SARS-CoV-2 nucleocapsid protein, as a marker for viral persistence, in skeletal muscle. Skeletal muscle biopsies from before the SARS-CoV-2 pandemic showed no signal when stained with polyclonal IgG anti-SARS-CoV-2 nucleocapsid protein, indicating that SARS-

CoV-2 nucleocapsid protein was not present in uninfected people which confirmed there was no aspecific binding. Continuous parametric longitudinal data (panel **A**) were analyzed with a generalized linear mixed model with a two-sided ANOVA. Bar: 100 μm. PEM post-exertional malaise. Box plots show the median (centerline), the first and third quartiles (the lower and upper bound of the box), and the whiskers show the 1.5× interquartile range. Source data are provided in the Source Data File.

exertional malaise. Furthermore, we did not observe more B-cell infiltration in the skeletal muscle of patients before and during post-exertional malaise. The mere presence of SARS-CoV-2 nucleocapsid protein in skeletal muscle is unsurprising, as nucleocapsid protein can be present up to a year after infection in blood[49,50]. It is however unknown if the full virus is present, or only protein remnants. Because the spike protein is located on the exterior of the virus, this protein may have different proinflammatory/coagulatory effects compared with the nucleocapsid protein. The absence of clear distinctions in the quantity of nucleocapsid protein and the equal presence of B- and T-cells following exercise suggests that factors other than viral persistence are associated with the pathophysiology of post-exertional malaise in patients with long COVID.

This study is observational in nature, and therefore we cannot establish causality. We only included individuals with severe long COVID, but the observed abnormalities are not reflective of physical inactivity. Accelerometer data indicated that the long COVID patients were not bed-ridden, and had an average step-count of ~4000 steps/ day (Supplemental Fig. 2C). While our cohort might have displayed sedentary behavior, physical inactivity itself does not cause post-exertional malaise and is linked with muscle atrophy and a lower mitochondrial SDH activity[51], which we did not observe in our cross-sectional analysis. As the sample size of this study is relatively small, our results require replication in other cohorts. In support of our work, peripheral alterations in mitochondrial metabolism[39] and myopathy[12,52] have recently been observed in patients with long COVID, but how post-exertional malaise alters skeletal muscle alterations requires further confirmation. Post-exertional malaise is a symptom that is specific for certain patient populations, including long COVID and myalgic encephalomyelitis/chronic fatigue syndrome (ME/CFS). While it is unlikely that the pathophysiology of post-exertional malaise is completely different, future work should highlight if the underlying

pathophysiology is similar in all patient populations, including children and adolescents[53].

In conclusion, this study reveals that local and systemic metabolic disturbances, severe exercise-induced myopathy, infiltration of amyloid-containing deposits, and immune cells in skeletal muscles of long COVID are key characteristics of post-exertional malaise. While these explain the symptomatology of post-exertional malaise in long COVID, the molecular pathways underlying these alterations in patients suffering from post-exertional malaise remain to be determined.

## Methods
### Study population
We performed a prospective case-control study in the post-COVID-19 clinic of the Amsterdam University Medical Centers (UMC) and the Faculty of Behavioral and Movement Sciences (Vrije Universiteit Amsterdam). The study protocol was approved by the medical ethics committee of the Amsterdam UMC (NL78394.018.21) and registered at www.clinicaltrials.gov (NCT05225688). All participants signed a written informed consent before participation. The study was conducted in accordance with the Declaration of Helsinki.

Long COVID patients were diagnosed by two experienced clinicians for long COVID symptomology and exclusion of potential differential diagnoses. All long COVID patients were diagnosed with post-exertional malaise (PEM) by the DSQ-PEM[54], had a minimum period of long COVID-related symptoms of six months, and were between 18 and 65 years old. No symptoms were present before the confirmed diagnosis of SARS-CoV-2. Questionnaires regarding fatigue, fatigue severity score (FSS)[55], multidimensional fatigue inventory (MFI)[56], and DSQ-PEM were obtained throughout the study period. None of the included participants were admitted to the hospital during acute SARS-CoV-2 infection and were healthy prior to NAAT or serology-proven SARS-

CoV-2 infection. Exclusion criteria were a medical history of cardio-vascular/pulmonary disease, diabetes mellitus, concurrent treatment with metabolism or coagulant-altering drugs during the study period (statins, corticosteroids, SGLT2 inhibitors, GLP1 receptor agonists, platelet aggregation blockers and any anticoagulants), adiposity but not BMI (limiting the biopsy of the musculus vastus lateralis), pregnancy, active infection, severe renal dysfunction or any other prior chronic illness known to impact clinical performance or >6 alcohol units per day or >14 alcohol units per week. Smoking was not an exclusion criteria.

None of the healthy controls had residual symptoms after the SARS-CoV-2 infection. One long COVID patient was excluded due to a recent SARS-CoV-2 re-infection (<7 days). Healthy controls withdrew because of the invasive nature of the protocol ($n = 2$), symptoms related to a burn-out ($n = 1$), and a novel diagnosis of uncontrolled hypertension ($n = 1$).

## Study design
All participants followed the same study protocol (Supplemental Fig. 1), with four visits over a two-week period. Participants arrived at the laboratory at least two hours postprandial and were instructed to avoid caffeine on the day of the laboratory visit. Post-exertional malaise was induced by a maximal incremental ramp exercise test on day 7. Skeletal muscle biopsies were taken from the vastus lateralis on day 1 (before) and day 8 (1 day after exercise testing) using a suction-supported Bergström needle (Pelomi, Albertlund, Denmark). Both biopsies were taken from the same ~2 cm incision; the first biopsy was taken towards the proximal end of the muscle and the second biopsy was taken towards the distal part of the muscle. Samples were stored at −195 °C until subsequent analysis, or stored as otherwise stated. Venous blood was drawn on every visit. Ethylenediaminetetraacetic acid (EDTA) anticoagulated blood was obtained at each visit, and plasma was stored at −70 °C within 4 h. The strobe checklist can be found in the supplementary data file.

## Incremental ramp exercise test
To assess exercise tolerance and to induce post-exertional malaise in long COVID patients, we conducted an incremental ramp exercise test on a cycle ergometer with simultaneous ECG, pulmonary gas exchange/ventilation and muscle deoxygenation measurements. All exercise tests were conducted on an electronically braked cycle ergometer (Lode Excalibur Sport, Lode, Groningen, The Netherlands). The test was preceded by 2-min quiet rest on the ergometer and 4-min baseline cycling at low intensity (between 0 and 20 W depending on height, body mass and anticipated fitness). This was followed by a ramped, linear increase in work rate until task failure. The individual baseline work rates and ramp slopes were selected based on each participant's anthropometric characteristics and physical activity levels and designed to elicit task failure within 8–12 min. Participants were instructed to maintain their cadence between 70 and 90 revolutions/min, and task failure was defined as the point at which cadence dropped <60 revolutions/min despite verbal encouragement. Capillary lactate concentrations were determined at rest prior to the onset of the test, during baseline cycling, and immediately following task failure (Lactate Pro 2 LT-1730, ARKRAY Ltd., United Kingdom). Pulmonary gas exchange and ventilation were measured on a breath-by-breath basis (Cosmed Quark CPET; Cosmed, Rome, Italy).

## Near-infrared spectroscopy (NIRS).
Participants wore a spatially-resolved, continuous-wavelength portable NIRS device (Portamon, Artinis Medical Systems, Arnhem, The Netherlands) on the surface of the vastus lateralis muscle, halfway between the head of the femur and lateral condyle. This was around the location of the muscle biopsy. Measurements for total (Δtotal[heme]), deoxygenated (Δdeoxy[heme]) and oxygenated [heme] (Δoxy[heme]) [hemoglobin +

myoglobin], as well as muscle saturation ($SO_2$, i.e., oxy[heme]/total[heme]·100) were normalized to a maximal and minimal value obtained during blood flow occlusion proximally to the NIRS device using an inflatable cuff. Briefly, a cuff was inflated between 300–500 mmHg, and remained inflated until a plateau in Δtotal[heme], Δdeoxy[heme], and Δoxy[heme] signals were reached, and then subsequently released. The maximum and minimum yielded the physiological range for each individual, and all values were normalized to this physiological range. As the greatest source-detector separation distance of the NIRS device was 40 mm, participants with an adipose tissue thickness ≥20 mm (or skinfold thickness ≥40 mm) were excluded from this analysis. This resulted in the inclusion of 18 controls and 16 patients in the final analysis. Individual Δdeoxy[heme] data were averaged as a function of absolute and relative power output. All NIRS data was analyzed related to both absolute and relative work rates. Absolute work rates were analyzed at 25 W increments from 50 W until 175 W, after which most participants could not continue the test. NIRS data was averaged for the 5 W below and above each increment to reduce noise in the data set. Similarly, for relative work rates, data was analyzed at 0%, 20%, 40%, 60%, 80%, and 100% of maximal work rate. Data was averaged for the 5% above and below each increment. All data was subsequently analyzed using a linear mixed model.

## Daily life physical activity level
General daily activity was measured by an accelerometer (Actigraph wGT3X-BT, 60 Hz sampling time) worn on the right hip (anterior axillary line) during the entire study period, as described previously[57]. Participants were instructed to wear the accelerometer from early morning until bedtime. Daily physical activity level was determined by the number of steps per day. The time at which participants went to sleep and woke up was registered to allow adjustment of cortisol measurements.

## Skeletal muscle measurements
**Skeletal muscle respirometry.** Mitochondrial function was assessed in permeabilized fibers as described previously[27]. Small bundles of freshly isolated fibers were permeabilized with 50 μg mL$^{-1}$ saponin for 20 min at 4 °C in a solution consisting of (in mM) CaEGTA (2.8), EGTA (7.2), ATP (5.8), MgCl$_2$ (6.6), taurine (20), phosphocreatine (15), imidazole (20), DTT (0.5) and MES (50) (pH 7.1). Tissue was washed in respiration solution containing EGTA (0.5), MgCl$_2$ (3), K-lactobionate (60), taurine (20), KH$_2$PO$_4$ (10), HEPES (20), sucrose (110) and 1 g L$^{-1}$ fatty acid-free BSA (pH 7.1), quickly blotted dry, weighed and transferred to a respirometer (Oxygraph-2k; Oroboros Instruments, Innsbruck, Austria) in respiration solution at 37 °C. Oxygen concentration was maintained above 300 μM throughout the experiment to avoid limitations in oxygen supply. Background respiration was assessed before adding substrates and was subtracted from all subsequent values. Leak respiration was assessed after the addition of sodium glutamate (10 mM), sodium malate (0.5 mM), and sodium pyruvate (5 mM). NADH (Complex I)-linked respiration was assessed after the addition of 5 mM ADP, and 10 μM cytochrome c to confirm the absence of outer-mitochondrial membrane damage. Maximal oxidative capacity, with simultaneous input via NADH and succinate-linked respiration, was measured after the addition of 10 mM succinate. Maximum uncoupled respiration was determined via titration of carbonylcyanide-4-trifluoro-methoxyphenylhydrazone (FCCP) in 0.5 μM steps until no further increase in oxygen consumption was observed. Succinate-linked respiration was measured after blocking mitochondrial complex I with 0.5 μM rotenone. Two measurements per sample were performed simultaneously, and the results were averaged. Respiration values were normalized to wet weight and expressed in pmol O$_2$·s$^{-1}$·mg$^{-1}$. Several ratios were subsequently calculated, namely the oxidative phosphorylation/electron transport system control ratio ($\frac{OxPhos}{ETS}$), the electron transport system/oxidative

**Table 3 | Details about antibodies, including dilutions and vendors**

| Parameter | Antibody, dilution, vendor | Secondary antibody (dilution), vendor | RRID |
|---|---|---|---|
| Muscle fiber type | BA-D5 (MHC-I), 1 µg/ml, DSHB<br>SC-71 (MHC-IIA), 1 µg/ml, DSHB<br>6H1 (MHC-IIX) 5 µg/ml, DSHB (60 min)<br>WGA 350, 1:25, TMO W11263 (30 min<br>MHCI (BA-D5), 1 µg/ml, DSHBMHC IIA (SC-71), 1 µg/ml, DSHB<br>MHC IIX (6H1), 5 µg/ml, DSHB (60 min)<br>WGA 350, 1:25, TMO W11263 (30 min) | Goat-anti-mouse IgG2b 555, 1:1000, Invitrogen A21147;<br>Goat anti-mouse IgG1 647, 1:1000, Invitrogen A21240;<br>Goat anti-mouse IgM 488, 1:1000, Invitrogen A21042 (60 min) | AB_2235587AB_2147165;<br>AB_1157897;<br>AB_2535783; AB_2535809;<br>AB_2535711 |
| Amyloid-containing deposits | Thioflavin T (50 µM stock), 1:10, Sigma-Aldrich (30 min)<br>WGA 555, 1:25, TMO W32464 (30 min) | N.A. | |
| Endothelial cells | CD31, IgG1 mouse, 1:50, Abcam ab9498 (overnight incubation)<br>WGA 350, 1:25, TMO W11263 (30 min) | Goat anti-mouse IgG1 647, 1:200, Invitrogen A21240 (60 min) | AB_726362 |
| Endothelial cells | Biotinylated UEA-1, 1:100,<br>B-1065, Vector Laboratories<br>(30 min) | Vectastain Elite ABC Kit PK-6100, Vector Laboratories (30 min);<br>ImmPACT™ AMEC Red Peroxidase Substrate SK-4285, Vector Laboratories (10 min) | AB_2336766<br>AB_2336819<br>AB_2336519 |
| Lymphatic vessel | LYVE1, IgG rabbit, 1:50 Abcam 10278 (overnight incubation)<br>WGA 350, 1:25, TMO W11263 (30 min) | Goat anti-rabbit IgG 647, 1:200, Invitrogen A32733 (60 min) | AB_881387 |
| SARS-CoV-2 nucleocapsid protein | SARS-CoV-2 Nucleocapsid Antibody, Rabbit PAb, Antigen Affinity Purified, 1:500; SinoBiological (60 min)<br>WGA 555, 1:25, TMO W32464 (30 min) | Alexa fluor goat-anti rabbit IgG 488 1:500, Invitrogen A27034 (60 min) | AB_2892769 |
| CD3 | SP7 (rabbit moab), 1:200, Thermo Sc. Rm-9107-S | | AB_149924 |
| CD68 | PG-M1 IgG 3, 1:200, DAKO M0876 | | AB_2074844 |
| CD20 | L26 IgG 2a, 1:1000,DAKO/M0755 | | AB_2282030 |

*RRID* research resource identifiers.

phosphorylation control efficiency ($\frac{ETS-OxPhos}{ETS}$), and the biochemical coupling efficiency ($\frac{ETS-Leak}{ETS}$)[58–60].

**Immuno-histochemistry.** To visualize muscle fiber morphology, signs of pathology, and the presence of amyloid-containing deposits and viral particles, (immuno-)histochemistry was carried out on 10 µm thick sections from the vastus lateralis muscle biopsies. Details about dilutions and vendors can be found in Table 1. Atrophy and necrosis in hematoxylin and eosin staining were evaluated by two pathologists blinded to the group allocation. Skeletal muscle infiltration of T-cells (CD3[+]), B-cells (CD20[+]) and macrophages (CD68[+]) was scored positively or negatively(Table 3)

**Fiber-type composition.** Skeletal muscle fiber type quantification and size were assessed using immunofluorescence techniques using primary antibodies against myosin heavy chain (MHC) I (BA-D5), MHC II (SC-71), and MHC IIx (6H1)[61]. A negative control, without a primary antibody, was included for background correction. Firstly, sections were air-dried for 10 min and then blocked with 10% normal goat serum (NGS) for 60 min. Subsequently, slides were washed 3 times for 5 min each in 1x phosphate-buffered saline (PBS) and incubated with the primary antibodies for 60 min at room temperature. Again, slides were washed and subsequently incubated with the secondary antibodies for 60 min in the dark at room temperature. Once more, a washing step was performed and subsequently incubation of the slides with Wheat Germ Agglutinin (WGA) for 30 minutes in the dark at room temperature, followed by a final washing step and mounting the sections with coverslips using Vectashield Vibrance. Analysis was performed with ImageJ and Sandia Matlab AnalysiS Hierarchy (SMASH) Toolbox (version 1.0) in Matlab (version 2022a). Fiber type distribution was expressed using two methods: first, by calculating the percentage of fibers expressing each MHC compared to the total number of fibers present in the muscle section. This allowed for differentiation between fibers expressing both MHC-I and MHC-IIa, MHC-IIa only, and both MHC-IIx and MHC-IIa. Second, fiber distribution was expressed as the cross-sectional area occupied by each fiber type as a percentage of the cross-sectional area of all fibers. This allowed for quantification of hypertrophy/atrophy that may be occurring with particular MHC expression.

**Succinate dehydrogenase activity.** Sections were stained for succinate dehydrogenase (SDH) activity, which was used as a proxy for mitochondrial density, as described previously[62]. Immediately after cutting, sections were air-dried for 15 min and then immersed in a preheated (37 °C) solution containing sodium phosphate buffer (0.1 M, pH 7.6), sodium succinate (0.2 M), sodium azide (14 mM) and tetranitro blue tetrazolium (TNBT, 0.55 mM, Sigma) for 20 min in the dark, and the reaction was stopped by brief HCl (0.01 M) exposure before sections were washed and mounted with glycerin-gelatin. Sections were stored at 4 °C and imaged within ten days. The averaged SDH activity was obtained by outlining individual skeletal muscle fibers and absorbance was assessed using ImageJ (NIH, Bethesda, USA). Values were expressed as $\Delta A_{660}$ per µm tissue thickness per second of staining time ($\Delta A_{660}\,\mu m^{-1}\,s^{-1}$).

**Capillarization.** Capillarization was assessed by staining for Ulex Europaeus Agglutinin 1 lectin (UEA-1)[63]. Sections were air-dried for 10 min and fixated in ice-cold acetone (−20 °C) for 15 min. Subsequently, slides were washed 3 times for 2 min in PBS and blocked with 1% bovine serum album for 30 min. Afterwards, slides were incubated with UEA-1 for 30 min at room temperature, followed by wash steps and incubation with Vectastain Elite ABC kit for 30 min at room temperature. After washing, incubation with Red Peroxidase substrate for 10 min at room temperature was performed, washed, and sections were mounted with glycerin-gelatin (heated at 37 °C). Capillary-to-fiber

ratio (C/F) and capillary density (CD) by capillaries per mm$^2$ muscle tissue were determined in ImageJ.

**Amyloid-containing deposits.** To visualize amyloid-containing deposits, skeletal muscle sections were exposed to 5 μM Thioflavin T (ThT). The muscle sections with the most amyloid-containing deposits were additionally stained with ThT in combination with a CD31-specific antibody to visualize endothelial cells and a LYVE-1-specific antibody to visualize lymph vessels. Firstly, sections were air dried for 10 min and then fixed in 4% paraformaldehyde (PFA) for 5 min only for the double staining. Subsequently, slides were washed 3 times for 5 min each in PBS Tween (PBST) and blocked with 10% NGS for 60 min. Then, again a washing step was completed, followed by incubation with the primary antibody (either CD31 or LYVE1) in 0.1% bovine serum albumin (BSA) in PBS overnight at 4 °C. A negative control, without a primary antibody, was included. Slides were washed in PBST and subsequently incubated with the secondary antibody for 60 min in the dark at room temperature. Subsequently, a washing step was performed and followed by incubation in ThT in 0.1% BSA in PBS for 30 min, followed by another washing step. Lastly, incubation of the slides with WGA for 30 min in the dark at room temperature, followed by a final washing step and mounting of the sections with coverslips using Vectashield Vibrance.

**SARS-CoV-2.** To determine viral presence in the muscle cells, the sections were stained for SARS-CoV-2[64]. Staining began with air drying the sections for 10 min, followed by a blocking step with 10% NGS for 60 min. A washing step consisting of 3 times 5-min intervals in 1× PBS followed, after which the sections were incubated with the primary antibody anti-N for 60 minutes at room temperature. Another washing step followed, and then incubation with the secondary antibody for 60 min in the dark at room temperature. Afterward, a washing step was again performed, followed by incubation with WGA for 30 min in the dark at room temperature. Sections were washed a final time and mounted with a coverslip using a Vectashield hard set with DAPI.

**Image acquisition.** Slides were dried overnight at 4 °C before imaging. Images for fiber type and capillarization analysis were taken at ×20 magnification with VS200 Research Slide Scanner (Olympus) using Slideview 5.0. SDH images were taken with a DMRB microscope (Leica, Wetzlar) using a CCD camera, calibrated gray filters, and an individual calibration curve at 660 nm. Images for the ThT combination staining were taken at ×40 magnification using the fluorescence microscope (Axiovert 200M; Zeiss) and image processing software Slidebook 5.0. Approximately fifteen images were taken per sample, with 5 images per slide-section. All images were manually evaluated to exclude those containing out-of-focus areas, artifacts, and large areas of connective tissue. Background correction for CD31 and LYVE-1 was done based on the negative control. For the ThT, background correction was performed by thresholding the negative control for 99.0%. All typical examples were deconvolved with Huygens Professional version 22.40 (Scientific Volume Imaging, The Netherlands, http://svi.nl), using the CMLE algorithm, with Acuity: −90, 90 iterations, and default settings for SNR and background values. For the ThT staining, a manual background correction was applied.

**Metabolomics.** Metabolomics was performed on skeletal muscle biopsies and venous blood samples as previously described, with minor adjustments[27,65]. Our metabolomics method is specifically tailored to analyze key polar metabolites of the most fundamental metabolic pathways with high confidence, instead of a broader untargeted method where identifications are more uncertain. Positive and negative ionization measurements were performed separately. In a 2 mL tube, containing either 20 μL of plasma or an average of 3.2 mg of freeze dried muscle tissue, the following amounts of internal standard dissolved in water were added to each sample: adenosine-$^{15}$N$_5$-monophosphate (5 nmol), adenosine-$^{15}$N$_5$-triphosphate (5 nmol), D$_4$-alanine (0.5 nmol), D$_7$-arginine (0.5 nmol), D$_3$-aspartic acid (0.5 nmol), D$_3$-carnitine (0.5 nmol), D$_4$-citric acid (0.5 nmol), $^{13}$C$_1$-citrulline (0.5 nmol), $^{13}$C$_6$-fructose-1,6-diphosphate (1 nmol), $^{13}$C$_2$-glycine (5 nmol), guanosine-$^{15}$N$_5$-monophosphate (5 nmol), guanosine-$^{15}$N$_5$-triphosphate (5 nmol), $^{13}$C$_6$-glucose (10 nmol), $^{13}$C$_6$-glucose-6-phosphate (1 nmol), D$_3$-glutamic acid (0.5 nmol), D$_5$-glutamine (0.5 nmol), D$_5$-glutathione (1 nmol), $^{13}$C$_6$-isoleucine (0.5 nmol), D$_3$-lactic acid (1 nmol), D$_3$-leucine (0.5 nmol), D$_4$-lysine (0.5 nmol), D$_3$-methionine (0.5 nmol), D$_6$-ornithine (0.5 nmol), D$_5$-phenylalanine (0.5 nmol), D$_7$-proline (0.5 nmol), $^{13}$C$_3$-pyruvate (0.5 nmol), D$_3$-serine (0.5 nmol), D$_6$-succinic acid (0.5 nmol), D$_4$-thymine (1 nmol), D$_5$-tryptophan (0.5 nmol), D$_4$-tyrosine (0.5 nmol), D$_8$-valine (0.5 nmol). Subsequently, solvents were added to achieve a total volume of 500 μL water, 500 μL methanol, and 1 mL chloroform. Muscle tissues were homogenized before the addition of chloroform using a Qiagen TissueLyser II for 5 min at 30 times/s with a 5 mm Qiagen Stainless Steel Bead in each tube. All samples were thoroughly mixed, before centrifugation for 10 min at 14,000 rpm.

The top layer, containing the polar phase, was transferred to a clean 1.5 mL tube and dried using a vacuum concentrator at 60 °C. Dried samples were reconstituted in 100 μL 6:4 (v/v) methanol:water. Metabolites were analyzed using a Waters Acquity ultra-high performance liquid chromatography system coupled to a Bruker Impact II™ Ultra-High Resolution Qq-Time-Of-Flight mass spectrometer. Samples were kept at 12 °C during analysis, and 5 μL of each sample was injected. Chromatographic separation was achieved using a Merck Millipore SeQuant ZIC-cHILIC column (PEEK 100 × 2.1 mm, 3 μm particle size). The column temperature was held at 30 °C. The mobile phase consisted of (A) 1:9 (v/v) acetonitrile:water and (B) 9:1 (v/v) acetonitrile:water, both containing 5 mmol/L ammonium acetate. Using a flow rate of 0.25 mL/min, the LC gradient consisted of: Dwell at 100% Solvent B, 0–2 min; Ramp to 54% Solvent B at 13.5 min; Ramp to 0% Solvent B at 13.51 min; Dwell at 0% Solvent B, 13.51–19 min; Ramp to 100% B at 19.01 min; Dwell at 100% Solvent B, 19.01–19.5 min. Column equilibration was done by increasing the flow rate to 0.4 mL/min at 100% B 19.5–21 min. MS data were acquired using negative and positive ionization in full scan mode over the range of $m/z$ 50–1200. Data were analyzed using Bruker TASQ software version 2.1.22.3. All reported metabolite intensities were normalized to either freeze-dried tissue weight or plasma volume, as well as to internal standards with comparable retention times and response in the MS. Metabolite identification has been based on a combination of accurate mass, (relative) retention times, ion mobility data and fragmentation spectra, compared to the analysis of a library of standards. Measurements below the lower limit of quantification were imputed as half the lowest value. Metabolites were excluded from analysis if they contained missing values for more than 5% of samples. Four samples from skeletal muscle metabolomics were excluded during quality control.

## Venous blood measurements

**Hormone concentrations.** Venous concentrations of cortisol, creatine kinase and creatine kinase-MB were measured in the Amsterdam UMC clinical laboratory, and details are provided in Table 4.

**Circulating mitochondrial DNA.** Mitochondrial DNA (mtDNA) measurements were performed using plasma samples. DNA was isolated according to the BIOKE NucleoSpin® Tissue kit. Briefly, plasma was separated from blood collected in vials coated with ethylenediaminetetraacetic acid by centrifuging at 1500×$g$ for 10 min. Subsequently, 50 μL of plasma was diluted in a solution containing 150 μL PBS, 25 μL proteinase K, and 200 μL Buffer B3. After incubation at 70 °C for 15 min, 210 μL of 96% ethanol was added to the solution and centrifuged for 1 min at 11,000×$g$, where the solution passed through a silica membrane column. The column was then placed in a new collection tube, washed with 500 μL Buffer BW and centrifuged for 1 min

**Table 4 | Details about methods for venous blood concentrations**

| Parameter | Equipment | Method | Reference limits |
|---|---|---|---|
| Cortisol | Roche cobas C8000 | Electro- chemilumines-cence immunoassay "ECLIA" Roche Diagnostics | 250–650 nmol/L |
| Creatine kinase | c702 Roche diagnostics | NAC-activated, 37 °C Roche Diagnostics | Male <171 U/L Female <145 U/L |
| Creatine kinase-MB | e602 Roche Diagnostics | Enzyme-labeled sandwich immunoassay Roche Diagnostics | <4 μg/L |

at 11,000×$g$. This was subsequently washed with 600 μL of buffer B5 and centrifuged again for 1 minute at 11,000×$g$. Lastly, 100 μL of buffer BE was added, incubated for 3 min, and centrifuged a final time for 1 min at 11,000×$g$ to yield the final isolated DNA in solution. Both ND1 and ND6 mtDNA primers were used to quantify the mtDNA copy number, using the ΔΔCt-method. Each well contained 2 μL of mtDNA, 3 μL of primer mix, and 5 μl of Power Up SYBR Green Master mix. Subsequently, qPCR in duplo was conducted using QuantStudio 3 real-time PCR to quantify concentrations of mtDNA relative to the nuclear reference. [18]S. Values were normalized to the first time point.

## Statistical analysis

Distributions were assessed with histograms and Shapiro–Wilk tests. Categorical values are noted in the absolute numbers with percentages in brackets. Parametric quantitative variables are presented as means ± standard deviation (SD), and nonparametric quantitative variables are presented as median and interquartile ranges (IQR; 25th and 75th percentiles). Boxplots are shown with the median and IQR (25th and 75th percentiles) unless otherwise specified. All individual points represent a participant.

Non-normally distributed data were transformed using Box-Cox transformation in order to obtain a normal distribution for statistical analysis[66]. Categorical data were analyzed using a Fisher's exact test. Continuous parametric data were analyzed using a $t$-test or analysis of variance, with Tukey HSD post-hoc testing, where appropriate. Continuous nonparametric data were analyzed using the Mann–Whitney $U$ test, Kruskal–Wallis $H$ test, or pairwise Kruskal–Wallis test with Benjamini–Hochberg (BH) correction where appropriate. Correlations were analyzed using the Pearson correlation coefficient for linear relationships and the Spearman rank correlation coefficient for non-linear relationships. Correlation coefficients were compared using the R package *cocor*[67]. Categorical longitudinal data were analyzed using logistic regression with Time (baseline and 1-day after exercise) as a covariate. Continuous parametric longitudinal data and metabolomics were analyzed with a generalized linear mixed model with time (baseline, 1 day, and 1 week after exercise), group and interaction effects. Metabolomics data were adjusted with BH correction within the pre-specified metabolome pathway. Post-hoc comparisons of longitudinal data were, if the interaction term was significant, performed using the Empirical Mean Differences using R package *emmeans*[68] methods with BH adjustment. Continuous nonparametric longitudinal data analysis was conducted by comparing the delta (Δ) between the time points. Longitudinal metabolomic data were visualized with the effect size calculated using Hedges' $g$[69]. $p$-values < 0.05 (two-tailed) were considered statistically significant. All data were analyzed using R studio built under R version 4.0.3 (R Core Team 2013, Vienna, Austria).

## Reporting summary

Further information on research design is available in the Nature Portfolio Reporting Summary linked to this article.

## Data availability

The source data file is available below. The raw metabolomics data generated in this study have been deposited in the data publication platform of the Vrije University Amsterdam, with https://doi.org/10.48338/VU01-KPABVO. Metabolomics is available on MetaboLights (with study identification code: MTBLS9103). Source data are provided with this paper.

## Code availability

R codes are available at https://doi.org/10.5281/zenodo.10171056.

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

## Acknowledgements

We particularly thank all participants for their time and dedication in participating in this study. We also would like to thank all staff of the post-COVID outpatient clinic of the Amsterdam UMC, in particular, Drs. Mooij-Kalverda and Huismans, for their support during this study. We also acknowledge the help of Dr. Eric Voorn in the data acquisition of the physical activity measurements, Onno van Driel in the analysis of the fiber type distribution, and Jos de Koning in supporting exercise data collection. This study was funded by own funding, the Patient-Led Research Collaborative for Long COVID (Grant ID: C1, [MvV]), the Talud Foundation for the Amsterdam UMC Corona Research Fund [WWJ], AMC Foundation [MvV], VU Foundation [RCIW], ZonMw Onderzoek-sprogramma ME/CVS [RCIW], and the 2022 Ramsay Grant Program [RCIW].

## Author contributions

B.A., B.C., M.v.V., and R.W. have made substantial contributions to the conception, design, financial acquisition, analysis, and interpretation of the data, drafting of the work, and revised the manuscript. R.G., T.K., E.B., W.N., and C.O. have made substantial contributions to the conception, design, analysis, and interpretation of the data, drafting of the work, and substantially revised the manuscript. F.B., M.v.W., B.S., P.C., J.P., E.A., and W.J.W. have made substantial contributions to the conception, design, analysis of the data, drafting, and revision of the manuscript. All authors have approved the submitted version and agree to be personally accountable for the author's own contributions and to ensure that questions related to the accuracy or integrity of any part of the work, even ones in which the author was not personally involved, are appropriately investigated, resolved, and the resolution documented in the literature.

## Competing interests

The authors declare no competing interests.
