## [Peer Review File · Nature Communications]

REVIEWER COMMENTS

Reviewer #1 (Remarks to the Author):

I thank the authors for this well written manuscript investigating multiple mechanisms for PEM in long COVID. The characterization of participants and study protocol is properly explained.

The major limitation of the study is the small sample size, which is understandable considering the extensive investigations, but it still leads to difficulties with power and spurious associations. In addition, multiple comparisons are made. This should be mentioned in the discussion together with a statement that the results of the study needs replication. Also, little attention is given to the fact that the cases and controls differ regarding their physical activity and how that might confound some of the results. The only statement is "Physical inactivity itself also

does not cause PEM, and factors contributing to exercise intolerance do not necessarily contribute to the pathophysiology of PEM." I would like to see something here on the response to physical activity in physical inactive versus exercise intolerance.

My main concern lies in the statistical methods, by and large I can make a guess as to what test is used for what but the description as currently written does not support replication of the study. Requesting the R code should not be necessary to understand the statistical methods. Considering the low number of observations the use of any parametric test here can be questioned, especially as they require transformation of variables. Report number of observations in each figure.

Considering the nature of the study including an intervention (induction of PEM) I wonder if it has been registered in a clinical trial registry prior to study start?

Reviewer #2 (Remarks to the Author):

This is an amazing and extremely important study which raises several important findings on Long Covid and PEM, with a potential to be translated in patients with other post viral syndromes.

About the methodology, I don't have major issues, probably except one. I appreciate the authors investigated the presence of the N protein, and the findings are quite impressive. Evidence of viral

persistence has been discussed recently, and this should be discussed also more in detail in the discussion (you can refer to PMID: 37385286). However, the great result would be to know if the presence of the Spike protein/antigens is different in the two groups. Can the authors investigate this? If not, at least this should be discussed and mentioned in the study limitation, as the S protein may have different pro inflammatory/coagulatory effects compared with the N.

The findings about amyloid are also important. I didn't fully understand if the authors also evaluated elements of microclots in the biopsy or abnormal endothelium? This would be important as may confirm or not the microclots theory. In particular the authors mention that patients have impaired "peripheral oxygen extraction", can this be due to endothelial abnormalities? Have been this investigated? If the authors only evaluated the presence of amyloid but not the endothelium, this should be clarified and the theory of chronic endothelitis of capillaries remains open. Please clarify

In my opinion in the discussion the authors can be braver. First, I think this is one of the greatest proof that LC is a real disease, as this is debated still in 2023, and particularly PEM and fatigue. Similar PEM and abnormal muscle dysfunction has also been found in two pediatric studies and the authors should therefore mention that similar data are plausible also in adolescents (if not children) (please refer to PMID: 37378466 PMID: 37097045)

In the discussion, maybe the authors can mention some possible mechanisms and hypothetical therapeutic strategies?

Reviewer #3 (Remarks to the Author):

This investigation of the mechanistic bases for Covid-19 post-exertional malaise is timely and valuable. Appelman et al. present some novel and exciting data supporting a skeletal muscle axis for exercise dysfunction. The design was opportunistic and commendable – though there are some methodological issues that need consideration and greater explanation of limitations.

Specific/Minor Comments

80 Jones et al. (1979) defined the impact/importance of considering breathing pattern and VT when relating PETCO₂ to arterial PCO₂. Was that done herein? If CO₂ production is reduced drastically as an effect of lowered exercise capacity and VO₂max but maximal exercise ventilation is less affected the result will be greater hyperventilation and higher RER. Whilst mixed expired PCO₂ can, via the Bohr equation intuit V/Q mismatch, due to the impact of RR and TV on PETCO₂, it is not ipso facto evidence of V/Q mismatch and increased dead space ventilation.

87 Is it possible that the lower “peripheral” O₂ extraction might simply be a reflection of the lower work rate and/or VO₂ (see Ferreira et al. 2006)? There may also be a reduced capacity to direct blood flow and O₂ delivery within skeletal muscle that reduces O₂ extraction as presented by Heinonen et al. (2015). See below comment regarding NIRS figure.

Jones NL et al. Difference between end-tidal and arterial PCO₂ in exercise. *J Appl Physiol Respir Environ Exerc Physiol*. 1979 Nov;47(5):954-60. doi: 10.1152/jappl.1979.47.5.954. PMID: 511720.

Ferreira LF et al. Blood flow and O₂ extraction as a function of O₂ uptake in muscles composed of different fiber types. *Respir Physiol Neurobiol*. 2006 Oct 27;153(3):237-49. doi: 10.1016/j.resp.2005.11.004. Epub 2005 Dec 22. PMID: 16376620.

Heinonen I et al. Heterogeneity of Muscle Blood Flow and Metabolism: Influence of Exercise, Aging, and Disease States. *Exerc Sport Sci Rev*. 2015 Jul;43(3):117-24. doi: 10.1249/JES.000000000000044. PMID: 25688763; PMCID: PMC4470710.

94 “adaptation” or pathology? Please clarify.

97,8 Really interesting and potentially important. Though use of hyperoxia during functional analyses weakens this evidence.

104 “upon” or “following”

110 Perhaps state “...in the long Covid patients but not in Controls???”

~130 Given the expertise among authors here it is to be hoped that they are assessing VO₂ kinetics.....perhaps for another venue?

146 In heart failure and also type 2 diabetes the greatest effects for O₂ transport may be found in the microcirculation and relate not to loss of capillaries but stoppage of RBC flux in a substantial proportion of capillaries (Richardson et al. 2002; Padilla et al. 2006). Might the same phenomenon be extant here? Notably, in those preclinical studies, capillaries were stopped without visible evidence of occlusion (qualify line 213?).

Richardson TE et al. Effects of chronic heart failure on skeletal muscle capillary hemodynamics at rest and during contractions. *J Appl Physiol* (1985). 2003 Sep;95(3):1055-62. doi: 10.1152/jappphysiol.00308.2003. Epub 2003 May 9. PMID: 12740313.

Padilla DJ et al. Effects of Type II diabetes on capillary hemodynamics in skeletal muscle. *Am J Physiol Heart Circ Physiol*. 2006 Nov;291(5):H2439-44. doi: 10.1152/ajpheart.00290.2006. Epub 2006 Jul 14. PMID: 16844923.

Was O₂ saturation measured during maximal exercise herein?

Figure 1A and Methods. Were these VO₂max’s validated as recommended? (Poole & Jones, 2017)

Poole DC, Jones AM. Measurement of the maximum oxygen uptake $\dot{V}O_{2max}$: $\dot{V}O_{2peak}$ is no longer acceptable. *J Appl Physiol* (1985). 2017 Apr 1;122(4):997-1002. doi: 10.1152/jappphysiol.01063.2016. Epub 2017 Feb 2. PMID: 28153947.

Please correct units in 1C (GET).

1D. If plotted as a function of absolute power will these relationships overly one another? If so, the meaning of this with respect to exercise intolerance mechanisms may have to be rethought.

Figure 2C. Is it possible to identify fiber types in this figure?

410 Sex?

Reviewer #4 (Remarks to the Author):

The study explores the causes of post-exertional malaise (PEM) and exercise intolerance in long COVID patients. It found that metabolic disturbances, muscle tissue damage, and amyloid deposits in muscles contribute to PEM, while exercise intolerance is due to more high-fatigable fibers and lower mitochondrial function. The study involved 25 long COVID patients and 21 recovered controls, showing exercise intolerance relates to muscle metabolic dysfunction, amyloid deposits, and tissue damage.

The experimental design and study were conducted from my examination. The manuscript was very well written and this research represents a significant contribution to the understanding of long COVID symptoms, in particular regarding PEM.

On the metabolomics component of the study:

> L585 and 586 - "Metabolomics were performed as previously described, with minor adjustments^{26,56}. The full

586 protocol can be found in the supplementary information."

L589-590 - The authors wrote "Metabolite identification has been based on a combination of accurate mass, (relative) retention times, ion mobility data and fragmentation spectra, compared to the analysis of a library of standards."

First of all, the impact HD II qtof doesn't have ion mobility capability so it is impossible that it was used for annotation. The reference 56 that describes in details the protocol confirms that and the reference 26 isn't related at all (an Orbitrap mass spectrometer was used there). Then it might be helpful for other researcher to explain that HILIC chromatography is designed for the most polar metabolites and that for this reason the metabolome coverage is not extensive (even if sufficient to observed changes in metabolic pathway). Also it isn't clear from the method description if the positive and negative

ionisation mode were used in the same run (alternating scans - not sure if the Maxis HD II can do that though) or different analysis.

More problematic in my opinion, is the fact the mass spectrometry data are not available (only upon request):

Ensuring adherence to the F.A.I.R principles (Findable, Accessible, Interoperable, Reusable) by the global scientific community is paramount, especially when it involves data critical to addressing public health crises such as the COVID-19 pandemic. I strongly encourage the authors to deposit their mass spectrometry data, including basic metadata, into public repositories like MetaboLights (<https://www.ebi.ac.uk/metabolights/>) or the MassIVE/GNPS (<https://massive.ucsd.edu/ProteoSAFe/static/massive.jsp>) platform that are recommended by Nature (<https://www.nature.com/sdata/policies/repositories>), and to reference the deposition numbers in the manuscript. There are great resources available to help in the deposition on both repositories. This practice is not only essential for facilitating research into long COVID but also invaluable for fostering collaboration and advancement within the broader scientific community. Moreover, metabolomics is at the center of the present article and not having the data available is a strong limitation to the impact and reproducibility of the research. Also it is now often made mandatory by many research agencies for reproducibility. If there is commercial reason for not depositing, this must be declared in conflict of interest.

REVIEWER COMMENTS

Reviewer #1 (Remarks to the Author):

Reviewer comment 1

I thank the authors for this well written manuscript investigating multiple mechanisms for PEM in long COVID. The characterization of participants and study protocol is properly explained.

Response: We thank the reviewer for his/her positive evaluation of our manuscript. We have listed below a point-to-point response to your comment and where we have adjusted this in the manuscript.

Reviewer comment 2

The major limitation of the study is the small sample size, which is understandable considering the extensive investigations, but it still leads to difficulties with power and spurious associations. In addition, multiple comparisons are made. This should be mentioned in the discussion together with a statement that the results of the study needs replication.

Response: We agree with the relatively low sample in our study (25 patients and 21 healthy controls), and agree that replication is needed. We have added this to the discussion. It should be mentioned however that inducing post-exertional malaise in many patients with Long COVID comes with an ethical concern, as our participants had lasting symptoms of the 15 minutes' CPET for more than two weeks. Also, our study was meant as the first explorative study on the skeletal muscle component of a lower exercise capacity and post-exertional malaise in patients with Long COVID.

Discussion, page 10, line 370-379: *"As the sample size of this study is relatively small, the results require replication in other cohorts. In support of our work, peripheral alterations in mitochondrial metabolism (Colosio 2023), and myopathy (Hejbol 2022) have been observed in patients with long COVID, but how post-exertional malaise alters skeletal muscle alterations requires further confirmation."*

Reviewer comment 3

Also, little attention is given to the fact that the cases and controls differ regarding their physical activity and how that might confound some of the results. The only statement is "Physical inactivity itself also does not cause PEM, and factors contributing to exercise intolerance do not necessarily contribute to the pathophysiology of PEM." I would like to see something here on the response to physical activity in physical inactive versus exercise intolerance.

Response: We have added the following text to the discussion.

Discussion, page 10, line 357-363: *"We only included individuals with severe long COVID, but the observed abnormalities are not reflective of physical inactivity. Accelerometer data indicated that long COVID patients were not bed-ridden, and had an average step-count of ~4000 steps/day (ED Fig.2C). While our cohort might display sedentary behavior, physical inactivity itself does not cause post-exertional malaise, and is linked with muscle atrophy and a lower mitochondrial SDH activity⁴⁸, which we did not observe in our cross-sectional analysis"*.

Reviewer comment 4

My main concern lies in the statistical methods, by and large I can make a guess as to what test is used for what but the description as currently written does not support replication of the study. Requesting the R code should not be necessary to understand the statistical

methods. Considering the low number of observations the use of any parametric test here can be questioned, especially as they require transformation of variables. Report number of observations in each figure.

Response: We have now included an expanded statistical section in the methods, have provided all statistics used at each figure as well as the number of observations and provided all source data/R Code required for reproduction of the figures/tables. Furthermore, all measurements with a single time point from Figure 1 were also analyzed with a non-parametric test, as per suggestion, which did not change any of the results.

Raw metabolomics data are available on: <https://doi.org/10.48338/VU01-KPABVO>. R Code is available at <https://github.com/bappelman-AmsterdamUMC/Muscle-PASC>. The source data for all figures is attached to the submission and will be published online.

Reviewer comment 5

Considering the nature of the study including an intervention (induction of PEM) I wonder if it has been registered in a clinical trial registry prior to study start?

Response: The trial was registered before the start of the study. We have now included the registry code:

Methods, page 12, line 393-395:: *The study protocol was approved by the medical ethics committee of the Amsterdam UMC (NL78394.018.21) and registered at clinicaltrials.gov (NCT05225688).*

Reviewer #2 (Remarks to the Author):

This is an amazing and extremely important study which raises several important findings on Long Covid and PEM, with a potential to be translated in patients with other post viral syndromes.

General response: We are delighted that this reviewer shares our excitement for this study.

Reviewer comment 1

About the methodology, I don't have major issues, probably except one. I appreciate the authors investigated the presence of the N protein, and the findings are quite impressive. Evidence of viral persistence has been discussed recently, and this should be discussed also more in detail in the discussion (you can refer to PMID: 37385286). However, the great result would be to know if the presence of the Spike protein/antigens is different in the two groups. Can the authors investigate this? If not, at least this should be discussed and mentioned in the study limitation, as the S protein may have different proinflammatory/coagulatory effects compared with the N.

Response: The reviewer raises an interesting question, as the Spike protein clearly has different sensitivities for immune cell activation. In this study, we specifically asked ourselves the question whether the viral persistence would contribute to the development of post-exertional malaise. We know from other work that the Spike protein can remain inside the body for a long period after infection, but this does not necessarily entail the complete virus. We reasoned that if viral persistence plays a

role, then the SARS-CoV-2 nucleocapsid protein, which is located inside the virus, should be at least present. We have added this to the manuscript:

Results on page 7, line 258-260: *“We reasoned that if viral persistence plays a role, then the SARS-CoV-2 nucleocapsid protein, which is located inside the virus, should be at least present.”*

We have adapted the discussion setting including recent evidence of viral persistence.

Discussion, page 9, line 339-341: *“Viral infections can alter mitochondrial function, and multiple studies have also shown that residual SARS-CoV-2 protein presence is associated with long COVID (Swank et al, 2023, Patterson et al, 2022, Buonsenso et al 2023)”*

Because we did not observe any group or time effects in the nucleocapsid protein, we prefer not to speculate about the different proinflammatory/coagulatory effects of Spike compared with the nucleocapsid protein. We are aware of the possible role the viral persistence plays in the pathophysiology of long COVID, but in this publication, we prefer to focus on the pathophysiology of post-exertional malaise. We are convinced that this deserves further study, but as there is no change in nucleocapsid protein after the induction of post-exertional malaise, the primary focus of this study, this is currently outside the scope of this manuscript.

To follow the reviewer’s suggestions, we have added the following to the

Discussion, page 10, line 348-351: *“It is however unknown if the full virus is present, or only protein remnants. Because the spike protein is located at the exterior of the virus, this protein may have different proinflammatory/coagulatory effects compared with the nucleocapsid protein.”*

Reviewer comment 2

The findings about amyloid are also important. I didn’t fully understand if the authors also evaluated elements of microclots in the biopsy for abnormal endothelium? This would be important as may confirm or not the microclots theory. In particular the authors mention that patients have impaired "peripheral oxygen extraction", can this be due to endothelial abnormalities? Have been this investigated? If the authors only evaluated the presence of amyloid but not the endothelium, this should be clarified and the theory of chronic endotheliitis of capillaries remains open. Please clarify

Response: We are aware of the work that other scientists have done on endothelial dysfunction in patients with long COVID. There is however not a single marker for endothelial abnormalities and endotheliitis, and as such, drawing conclusions from the results of one particular marker is, in our opinion, inappropriate. Such a study requires a more elaborative approach from different angles, and as such, we reason that studying endothelial dysfunction in long COVID and its potential link with exercise intolerance and the development of PEM requires multiple new experiments. This would, in our view, slow down the current publication specifically targeting skeletal muscle aberrations and post-exertional malaise. We have added this avenue for future research to the discussion, as the reviewer suggests.

Discussion, page 9, line 323-326: *“We only evaluated the presence of amyloid, but not the function of the endothelium, or local blood flow. As such, the possible*

contribution of chronic endotheliitis or reduced blood flow to the development of post-exertional malaise and/or exercise intolerance in long COVID remains open (Charfeddine et al, 2021, Turner et al, 2023)”

Reviewer comment 3

In my opinion in the discussion the authors can be braver. First, I think this is one of the greatest proof that LC is a real disease, as this is debated still in 2023, and particularly PEM and fatigue. Similar PEM and abnormal muscle dysfunction has also been found in two pediatric studies and the authors should therefore mention that similar data are plausible also in adolescents (if not children) (please refer to PMID: 37378466 PMID: 37097045).

Response: We thank the reviewer for this opportunity. We have re-written the discussion, and added additional arguments related to the pathophysiology of long COVID, including adolescents, within the scope of the presented results.

In response to the reviewer comments, we have added the following sentence about children and adolescents.

Discussion, page 10, line 374-379: “Post-exertional malaise is a symptom that is specific for certain patient populations, including long COVID and myalgic encephalomyelitis/chronic fatigue syndrome (ME/CFS). While it is unlikely that the pathophysiology of post-exertional malaise is completely different, future work should highlight if the underlying pathophysiology is similar in all patient populations, including children and adolescents (Pizzuto et al, 2023).”

Reviewer comment 4

In the discussion, maybe the authors can mention some possible mechanisms and hypothetic therapeutic strategies?

Response: We prefer not to mention or speculate too much on treatment options in the discussion of our manuscript, without proper scientific support and argument. We have noticed that papers on the pathophysiology of long COVID attract a lot of attention from patients and colleague-scientists. To avoid any speculation and false hopes to patients, we prefer to be conservative about hypothetical therapeutic strategies.

We have, however, expanded the discussion regarding possible pathophysiological causes, as suggested by the reviewer.

Discussion, page 8-9, line 301-314: “We observed that a fiber-type shift and lower mitochondrial respiration contribute to the reduced exercise capacity in patients, but these do not necessarily contribute to the pathophysiology of post-exertional malaise, as fiber-type shifts occur at a very slow pace and could have been present before long COVID. Our participants showed a large variability in exercise tolerance (\$\dot{V}O_{2max}\$ and peak power), fiber type composition, and mitochondrial enzyme activity, but despite this large heterogeneity, all patients experienced post-exertional malaise one day after the exercise test. As such, we conclude that the pathophysiology of fatigue and a reduced exercise capacity is distinct from the rapid development of post-exertional malaise in long COVID patients. The development of post-exertional malaise could in turn, however, lead to a further reduction in exercise capacity in patients, as the acute reduction in mitochondrial SDH activity, occurrence of tissue necrosis and possibly intramuscular accumulation of amyloid-containing deposits

could worsen skeletal muscle metabolism and force production over time, causing a vicious downward circle.

Discussion, page 9, line 323-326: “We only evaluated the presence of amyloid, but not the function of the endothelium, or local blood flow. As such, the possible contribution of chronic endotheliitis or reduced blood flow to the development of post-exertional malaise and/or exercise intolerance in long COVID remains open (Charfeddine et al, 2021, Turner et al, 2023)”

Discussion, page 10, line 374-349: “Post-exertional malaise is a symptom that is specific for certain patient populations, including long COVID and myalgic encephalomyelitis/chronic fatigue syndrome (ME/CFS). While it is unlikely that the pathophysiology of post-exertional malaise is completely different, future work should highlight if the underlying pathophysiology is similar in all patient populations, including children and adolescents (Pizzuto et al, 2023).”

Reviewer #3 (Remarks to the Author):

This investigation of the mechanistic bases for Covid-19 post-exertional malaise is timely and valuable. Appelman et al. present some novel and exciting data supporting a skeletal muscle axis for exercise dysfunction. The design was opportunistic and commendable – though there are some methodological issues that need consideration and greater explanation of limitations.

Reviewer comment 1 - Specific/Minor Comments

80 Jones et al. (1979) defined the impact/importance of considering breathing pattern and VT when relating PETCO₂ to arterial PCO₂. Was that done herein? If CO₂ production is reduced drastically as an effect of lowered exercise capacity and VO₂max but maximal exercise ventilation is less affected the result will be greater hyperventilation and higher RER. Whilst mixed expired PCO₂ can, via the Bohr equation intuit V/Q mismatch, due to the impact of RR and TV on PETCO₂, it is not ipso facto evidence of V/Q mismatch and increased dead space ventilation.

Response: The reviewer raises a good point. To avoid over-interpretation of the data here, we have modified the passage to read as follows:

Results, page 3, line 93-98: “~~During exercise, Patients had lower maximal ventilation and lower maximal end-tidal partial pressure of CO₂ (PETCO₂, reflective of arterial PCO₂) implying poorer ventilatory function, and exaggerated hyperventilation with respect to CO₂ production, indicative of an altered matching of ventilation-to-perfusion and/or increased physiologic dead space during exercise in patients with long COVID (ED Fig.3A-C) (Frésard et al, 2022, Schwendinger et al, 2023, Skjørten et al, 2021. “~~

Reviewer comment 2: Is it possible that the lower “peripheral” O₂ extraction might simply be a reflection of the lower work rate and/or VO₂ (see Ferreira et al. 2006)? There may also be a reduced capacity to direct blood flow and O₂ delivery within skeletal muscle that reduces O₂ extraction as presented by Heinonen et al. (2015). See below comment regarding NIRS figure.

Jones NL et al. Difference between end-tidal and arterial PCO₂ in exercise. J Appl Physiol Respir Environ Exerc Physiol. 1979 Nov;47(5):954-60. doi: 10.1152/jappl.1979.47.5.954. PMID: 511720.

Ferreira LF et al. Blood flow and O₂ extraction as a function of O₂ uptake in muscles composed of different fiber types. Respir Physiol Neurobiol. 2006 Oct 27;153(3):237-49. doi: 10.1016/j.resp.2005.11.004. Epub 2005 Dec 22. PMID: 16376620.

Heinonen I et al. Heterogeneity of Muscle Blood Flow and Metabolism: Influence of Exercise, Aging, and Disease States. *Exerc Sport Sci Rev.* 2015 Jul;43(3):117-24. doi: 10.1249/JES.0000000000000044. PMID: 25688763; PMCID: PMC4470710.

Response: We agree with the reviewer that for accurate interpretation of the data it is necessary to present them both as a function of relative and absolute power output. We initially only displayed the values as a function of relative power output because we wanted to display this as part of their relative maximum. The results of this analysis are graphically summarized below, and are added to the manuscript in Figure 1D-E.

Long COVID patients had a significantly lower Δ deoxy[heme] but with a similar exercise-induced increase compared to healthy controls when calculated with the relative work rate.

The absolute work rate showed a plateau phase at 175W for long COVID patients with a borderline non-significant interaction term. This finding implies that, for any given external power output, patients with long COVID displayed a *reduced* muscle O_2 extraction.

We have added the following sections/figures to the Methods and Results.

Results, page 4, line 100-104: “The lower maximal O_2 pulse (i.e. the product of stroke volume and arteriovenous O_2 difference; ED Fig.3F), gas exchange threshold (Fig.1C) and peripheral O_2 extraction (determined via near-infrared spectroscopy; Fig.1D-E, ED Fig.3H) during exercise all indicated peripheral skeletal muscle impairments in patients.”

Figures, page 24, line 798-800: “D-E: Muscle deoxygenated [heme] responses (mean \pm SD) measured by near-infrared spectroscopy were lower in long COVID (n=16), indicative of reduced peripheral oxygen extraction during exercise as compared to healthy controls (n=18).”

We also expanded the method section regarding the NIRS. This also includes specific changes for the absolute power output.

Methods, page 13-14, line 446-466: *“Participants wore a spatially-resolved, continuous-wavelength portable NIRS device (Portamon, Artinis Medical Systems, Arnhem, The Netherlands) on the surface of the vastus lateralis muscle, halfway between head of the femur and lateral condyle. This was around the location of the muscle biopsy. Measurements for total (Δ total[heme]), deoxygenated (Δ deoxy[heme]) and oxygenated [heme] (Δ oxy[heme]) [hemoglobin + myoglobin], as well as muscle saturation (SO_2 , i.e., $\text{oxy[heme]}/\text{total[heme]} \cdot 100$) were normalized to a maximal and minimal value obtained during blood flow occlusion proximally to the NIRS device using an inflatable cuff. Briefly, a cuff was inflated between 300-500mmHg, and remained inflated until a plateau in Δ total[heme], Δ deoxy[heme], and Δ oxy[heme] signals were reached, and then subsequently released. The maximum and minimum yielded the physiological range for each individual, and all values were normalized to this physiological range. As the greatest source-detector separation distance of the NIRS device was 40 mm, participants with an adipose tissue thickness ≥ 20 mm (or skinfold thickness ≥ 40 mm) were excluded from this analysis. This resulted in the inclusion of 18 controls and 16 patients in the final analysis. Individual Δ deoxy[heme] data were averaged as a function of absolute and relative power output. All NIRS data was analyzed related to both absolute and relative work rates. Absolute work rates were analyzed at 25W increments from 50W to 175W, after which most patients could not continue the test. NIRS data was also analyzed at 0%, 20%, 40%, 60%, 80%, and 100% of maximal work rate. All data was subsequently analyzed using a linear mixed model.”*

Methods, page 13, line 460-462: *“Individual Δ deoxy[heme] data were averaged as a function of absolute and relative power output.”*

Reviewer comment 4:

110-112 “adaptation” or pathology? Please clarify.

Response: We agree with the reviewer that this may cause confusion. It is unknown whether this concerns an adaptation or pathology, and hence we have changed this phrase.

Results, page 4, line 119-121: *“For a given skeletal muscle size, patients did not attain the same peak power output, suggesting that exercise performance-intolerance in patients was at least partly explained by intrinsic alterations in skeletal muscle force and fatigue characteristics adaptations.”*

Reviewer comment 5

97,8 Really interesting and potentially important. Though use of hyperoxia during functional analyses weakens this evidence.

Response: We agree that this avoids the possible contribution of limiting O_2 tensions *in vivo*. For the correct determination of maximal mitochondrial oxidative phosphorylation capacity in permeabilized muscle fibers, it is critical to maintain O_2 tension at hyperoxic values in order to avoid diffusion limitation for O_2 . This is common practice for this measurement, as otherwise a lower mitochondrial respiration can simply be explained by a low PO_2 and diffusion limitation. Our respirometry findings indicate that the intrinsic function of the mitochondria of long COVID patients was impaired, irrespective of the potential (yet unconfirmed) influence

of compromised O₂ delivery during exercise performed *in vivo*. We have altered this passage to read as follows:

Results, page 5, line 142-143: *“Mitochondrial respiration was assessed in hyperoxic conditions to avoid diffusion limitation for oxygen that could contribute to exercise intolerance in vivo.”*

Reviewer comment 6

124 “upon” or “following”.

Response: We have adjusted the sentence.

Results, page 4, line 134-136: “All long COVID patients experienced post-exertional malaise following upon the exhaustive maximal exercise, despite the heterogeneity in exercise tolerance.”

Reviewer comment 7

110 Perhaps state “...in the long Covid patients but not in Controls???”

Response: Oxidative phosphorylation capacity was reduced in both groups one day following exercise, whereas SDH activity was only reduced in the long COVID group following the exercise test. We have now modified this passage of text to read as follows (page 6, line 134):

Results, page 4, line 146-151: *“Oxidative phosphorylation capacity decreased one day following the maximal exercise in both controls and patients (Fig.3A). SDH activity was not reduced in healthy controls one day after exercise, but was reduced in the long COVID patients, suggesting that the combination of a reduced maximal mitochondrial respiration and decreased mitochondrial content are part of the pathophysiology of post-exertional malaise.”*

Reviewer comment 8

~130 Given the expertise among authors here it is to be hoped that they are assessing VO₂ kinetics.....perhaps for another venue?

Response: Long COVID patients experience post-exertional malaise even upon moderate constant load exercise, and reliable VO₂ kinetics analysis requires multiple constant load exercise bouts on separate days. Additionally, VO₂ kinetics typically do not give specific insight into the location of the limitation (oxygen delivery and/or utilization rates). However, we assessed these directly in our muscle biopsies. We therefore do not anticipate performing such experiments in this patient population. Colosio et al. (37675472) recently showed evidence of slowed VO₂ kinetics in this patient cohort, and concluded that ‘limited exercise tolerance mainly due to “peripheral” determinants’. Their study and ours actually confirm this. We have added this to the manuscript:

Discussion, page 10, line 372-374: *“In support of our work, peripheral alterations in mitochondrial metabolism (Colosio et al. 2023), and myopathy (Hejbol 2022) have been observed in patients with long COVID, but how post-exertional malaise alters skeletal muscle alterations requires further confirmation.”*

Reviewer comment 9

In heart failure and also type 2 diabetes the greatest effects for O₂ transport may be found in

the microcirculation and relate not to loss of capillaries but stoppage of RBC flux in a substantial proportion of capillaries (Richardson et al. 2002; Padilla et al. 2006). Might the same phenomenon be extant here? Notably, in those preclinical studies, capillaries were stopped without visible evidence of occlusion (qualify line 213?).

Response: We thank the reviewer for this interesting hypothesis.

In this article, we tested the notion that accumulation of amyloid-containing deposits inside capillaries causes local hypoxia by local stoppage of blood flow. This has been hypothesized to be the cause of the diverse symptoms of long COVID (Kell et al. 2022, Aschman et al, 2023), and this was what we intended to address here.

We agree with the reviewer that a halted RBC flux is a possible reason for a potentially reduced blood flow. Unfortunately, we cannot directly measure microvascular blood flow, and do not know if local blood flow is reduced. The measurements suggested by the reviewer are presently intractable in humans and unfortunately, thus not possible for us to verify in this cohort. No animal model of Long COVID currently exists, to verify this hypothesis.

As opposed to the hypothesis, our NIRS-derived muscle deoxygenation data does not support the theory that a reduced microvascular blood flow and lower local O₂ delivery causes hypo-circulation in skeletal muscle during exercise, although we realize these data are not conclusive.

There are various other potential mechanisms that can explain a potentially reduced O₂ transport in patients. Since we do not find proof of an impaired (or unimpaired) blood flow, we prefer not to speculate about possible ways by which (local) blood flow can (or cannot) be impaired in long COVID.

In light of the reviewer suggestions, we have now revised the results section to include the following revised text:

Results, page 6, line 205-210: “Neither did we observe any signs of skeletal muscle tissue hypoxia, as the skeletal muscle capillary-to-fiber ratio, capillary density (Fig.2B) and intracellular and circulating lactate concentrations (ED Fig.3G+6A) were not different between long COVID patients and controls. Therefore, we conclude that post-exertional malaise cannot be explained by the hypothesis that these deposits block vessel perfusion, causing local tissue hypoxia (Kell et al. 2022, Aschman et al, 2023).”

And also within the limitations section of the discussion, to highlight that functional impairments may still persist:

Discussion, page 9, line 323-326: “We only evaluated the presence of amyloid, but not the function of the endothelium, or local blood flow. As such, the possible contribution of chronic endotheliitis or reduced blood flow to the development of post-exertional malaise and/or exercise intolerance in long COVID remains open (Charfeddine et al, 2021, Turner et al, 2023)”

Reviewer comment 10

Was O₂ saturation measured during maximal exercise herein?

Response: We assume that the reviewer refers to systemic O₂ saturation (i.e. SpO₂) not tissue O₂ saturation, which was assessed by NIRS and reported in the manuscript. We measured SpO₂ during rest during all hospital visits, but did not observe any

abnormalities. We did not assess SpO₂ during exercise. There were however no differences in end-tidal PO₂ at maximal exercise.

Reviewer comment 11

Figure 1A and Methods. Were these VO₂max's validated as recommended? (Poole & Jones, 2017).

Poole DC, Jones AM. Measurement of the maximum oxygen uptake $\dot{V}O_{2max}$: $\dot{V}O_{2peak}$ is no longer acceptable. J Appl Physiol (1985). 2017 Apr 1;122(4):997-1002. doi: 10.1152/jappphysiol.01063.2016. Epub 2017 Feb 2. PMID: 28153947.

Response: We did not perform a verification bout to validate our $\dot{V}O_{2max}$ estimates, simply because we were afraid of the additional PEM that followed the maximal exercise test. Given the fact that the PEM response to exercise was so severe and rapid in some of these patients, we feel it would not have been ethical to perform any further exercise after exhausting patients during the incremental test. Post-hoc analysis gave us confidence that our results do indeed represent a truly lower $\dot{V}O_{2max}$. Firstly, the proportion of participants attaining a plateau in $\dot{V}O_2$ did not differ between groups towards the end of the exercise test, which is the primary criterion that a true $\dot{V}O_{2max}$ has been attained. Secondly, the gas exchange threshold and respiratory compensation point occurred at the same percentage of $\dot{V}O_{2max}$ in the patients as in the controls. As such, it is highly unlikely that participants were making a submaximal effort in these tests. Third, although their use has been rightly cautioned (Poole&Jones 2017, 28153947), all secondary criteria ([lactate], RER, HR) were attained in both groups. We have added the following sentence:

Results, page 9, line 104-108: *"The lower $\dot{V}O_{2max}$ in patients was not due to submaximal effort during the exercise test, as the proportion of participants attaining a plateau in $\dot{V}O_2$ did not differ between groups, the gas exchange threshold was at similar percentages of $\dot{V}O_{2max}$, and secondary criteria ([lactate], maximal RER and heart rate) were attained in both groups (ED Fig.3E, G-H)."*

Reviewer comment 12

Please correct units in 1C (GET).

Response: This is corrected.

Reviewer comment 13

1D. If plotted as a function of absolute power will these relationships overly one another? If so, the meaning of this with respect to exercise intolerance mechanisms may have to be rethought.

Response: Please see our response to comment 2.

Reviewer comment 14

Figure 2C. Is it possible to identify fiber types in this figure?

Response: We have added the fiber types to figure 3C.

Reviewer comment 15

410 Sex?

Response: There is no mention of "sex" in line 410, and we assume a typo of the reviewer. We remain open to suggestions.

Reviewer #4 (Remarks to the Author):

The study explores the causes of post-exertional malaise (PEM) and exercise intolerance in long COVID patients. It found that metabolic disturbances, muscle tissue damage, and amyloid deposits in muscles contribute to PEM, while exercise intolerance is due to more high-fatigable fibers and lower mitochondrial function. The study involved 25 long COVID patients and 21 recovered controls, showing exercise intolerance relates to muscle metabolic dysfunction, amyloid deposits, and tissue damage.

The experimental design and study were conducted from my examination. The manuscript was very well written and this research represents a significant contribution to the understanding of long COVID symptoms, in particular regarding PEM.

General response: We thank the reviewer for the kind words and appreciation of the article.

Reviewer comment 1

On the metabolomics component of the study:

> L585 and 586 – “Metabolomics were performed as previously described, with minor adjustments^{26,56}. The full 586 protocol can be found in the supplementary information.”

L589-590 – The authors wrote “Metabolite identification has been based on a combination of accurate mass, (relative) retention times, ion mobility data and fragmentation spectra, compared to the analysis of a library of standards.”

First of all, the impact HD II qtof doesn't have ion mobility capability so it is impossible that it was used for annotation. The reference 56 that describes in details the protocol confirms that and the reference 26 isn't related at all (an Orbitrap mass spectrometer was used there). Then it might be helpful for other researcher to explain that HILIC chromatography is designed for the most polar metabolites and that for this reason the metabolome coverage is not extensive (even if sufficient to observed changes in metabolic pathway). Also it isn't clear from the method description if the positive and negative 1 ionization mode were used in the same run (alternating scans – not sure if the Maxis HD II can do that though) or different analysis.

Response: We thank the reviewer for the thorough feedback.

Indeed, the Impact II does not have ion mobility capabilities. Additionally, it should be mentioned that its fragmentation capabilities have also not been utilized directly in this manuscript. However, metabolite identification during the development stages of this method has been performed with a mixture of these techniques at the Core Facility Metabolomics at the AUMC (and continues to be performed). Additionally, metabolite identification in polar metabolomics is a highly pragmatic process that also contains steps such as the analysis of sample matrices with known defects, or peaks that we have determined to be caused by in-source fragmentation from other metabolites. On the other hand, when dealing with small metabolites, it is often not possible to perform MS/MS analysis, as their fragments are not informative or too small to detect. Still, we can have high confidence in them due to for instance the presence of an internal standard or lack of probable isomers. Because of this, we provide a more holistic confidence level in any metabolite's identification; provided in the comprehensive protocol as a reference. (Schomakers, B. V. et al. Polar metabolomics in human muscle biopsies using a liquid-liquid extraction and full-scan LC-MS. STAR Protoc 3, 101302 (2022)).

The text in the materials and methods should be viewed as a practical overview of the general strategies that were employed while developing the metabolomics method.

Furthermore, we now published all raw data in a separate DOI, allowing for full transparency. We have added the following sentence:

Supplemental methods, page 13: *Our metabolomics method is specifically tailored to analyze key polar metabolites of the most fundamental metabolic pathways with high confidence, instead of a broader untargeted method where identifications are more uncertain. Positive and negative ionization measurements were performed separately.*

More problematic in my opinion, is the fact the mass spectrometry data are not available (only upon request):

Ensuring adherence to the F.A.I.R principles (Findable, Accessible, Interoperable, Reusable) by the global scientific community is paramount, especially when it involves data critical to addressing public health crises such as the COVID-19 pandemic. I strongly encourage the authors to deposit their mass spectrometry data, including basic metadata, into public repositories like MetaboLights (<https://www.ebi.ac.uk/metabolights/>) or the MassIVE/GNPS (<https://massive.ucsd.edu/ProteoSAFe/static/massive.jsp>) platform that are recommended by Nature (<https://www.nature.com/sdata/policies/repositories>), and to reference the deposition numbers in the manuscript. There are great resources available to help in the deposition on both repositories. This practice is not only essential for facilitating research into long COVID but also invaluable for fostering collaboration and advancement within the broader scientific community. Moreover, metabolomics is at the center of the present article and not having the data available is a strong limitation to the impact and reproducibility of the research. Also it is now often made mandatory by many research agencies for reproducibility. If there is commercial reason for not depositing, this must be declared in conflict of interest.

Response: In accordance with the F.A.I.R. principles, we have uploaded all data related to the metabolomics in an online repository: DOI: <https://doi.org/10.48338/VU01-KPABVO> In these files, the key MS parameters (instrument type, mass resolution, ionization type, fragmentation type and energy), LC parameters (column, elution buffers, injection volume, gradient and time) can be found, as well as all the raw data.

REVIEWERS' COMMENTS

Reviewer #2 (Remarks to the Author):

The authors have, in my opinion, replied to all raised comments.

Good luck and thank you

Reviewer #3 (Remarks to the Author):

The authors have addressed the principal concerns of this reviewer and the revised manuscript has greater clarity and perceived impact. I believe that this work makes a substantial original contribution to this field.

Reviewer #4 (Remarks to the Author):

I am glad the authors decided to make the mass spectrometry data available (instead of available upon request). Still there are two issues before it is truly F.A.I.R compliant:

1) The MS data are not deposited to a suitable/recognized repository with the required amount of standardized metadata. They are presently deposited to an institutional generic repository which limits community-wide reuse/contextualization due to the lack of integration and findability in dedicated environment. Like I mentioned those are: MetaboLights, GNPS/MassIVE or MetabolomicsWorkbench ...

2) The authors deposited the mass spectrometry files in their vendor file format (.d from Bruker). The community typically provides the MS data in open-file format like mzML to facilitate reuse. This is especially critical for .d from Bruker as to my knowledge the conversion must be done with the Bruker proprietary software Data Analysis (commercial licence and only Windows system) because it requires application of lockmass correction (due to the internal lockmass).

Although the authors clarified some aspects of the MS methodology in the response and the manuscript, but warning some are untracked changes: Like the following sentence: L539-541[Our metabolomics method is specifically tailored to analyze key polar metabolites of the most fundamental metabolic pathways with high confidence, instead of a broader untargeted method where identifications are more uncertain.].

I still found unclear the explanations and description of the relationships between the actual MS data and analysis (that didn't have fragmentation spectra and ion mobility) and L545-546 "the Metabolite identification has been based on a combination of accurate mass, (relative) retention times, ion mobility data and fragmentation spectra, compared to the analysis of a library of standards". From the discussion, it seems that only the MS1 ion accurate mass and (relative?!) retention times were in fact used for the annotation of the present MS data, by matching them with those of internal standards from a library previously established (in other experimental conditions). Maybe what the authors are trying to indicate is that they previously analyzed the same biological matrix and identified the compounds using other experimental parameters (ion mobility and MS/MS) which indeed increase the confidence in the annotation. In any case, the MS acquisition and annotation method must be described more clearly in the manuscript.

Reviewer #5 (Remarks to the Author):

In general, the prior comments have been adequately addressed. The statistical analyses are now clearly explained and more reproducible. However, although the authors have now added a clinicaltrials.gov identifier, it seems the study was registered after the actual start date. If this is indeed the case, I would suggest that the authors clearly state in the manuscript that the study was retrospectively registered and include the dates of first study registration and first patient inclusion.

Point by point rebuttal

REVIEWERS' COMMENTS

Reviewer #4 (Remarks to the Author):

I am glad the authors decided to make the mass spectrometry data available (instead of available upon request). Still there are two issues before it is truly F.A.I.R compliant:

- 1) The MS data are not deposited to a suitable/recognized repository with the required amount of standardized metadata. They are presently deposited to an institutional generic repository which limits community-wide reuse/contextualization due to the lack of integration and findability in dedicated environment. Like I mentioned those are: MetaboLights, GNPS/MassIVE or MetabolomicsWorkbench ...

Response: Like the reviewer, we encourage FAIR principles, and have chosen for immediate release of our raw data, including metadata, which provided an immediate DOI. We prefer not to upload all our raw -omics data on a US-based open access server (MetabolomicsWorkbench owned by NIH), because data transfer outside the EU requires that our university (hospital) privacy and legal specialists to agree, and we did not state this US-data transfer specifically in our patient information sheet. We would therefore prefer to upload this on a European server, MetaboLights, but this repository currently has an expected waiting time of around one year before it is fully available to the public. We will process our data on this repository, but we feel that science should not be held back by waiting times. In the meantime, our dataset is still publicly available: <https://doi.org/10.48338/VU01-KPABVO>.

- 2) The authors deposited the mass spectrometry files in their vendor file format (.d from Bruker). The community typically provides the MS data in open-file format like mzML to facilitate reuse. This is especially critical for .d from Bruker as to my knowledge the conversion must be done with the Bruker proprietary software Data Analysis (commercial licence and only Windows system) because it requires application of lockmass correction (due to the internal lockmass).

Response: We appreciate the reviewer's comment, but there is no concern in this matter. Firstly, many uploaded files in MetaboLights, GNPS/MassIVE or MetabolomicsWorkbench are with the .d-file format, and as such, this is a standard format in the field. Secondly, the proteowizard software (freeware - <https://proteowizard.sourceforge.io/download.html>) can convert all files to MzML files, but there is a major point of attention: you can adjust file parameters, and delete metadata, such as the LC method and MS method. Therefore, there is no Bruker proprietary software needed. Thirdly, the raw data also contains time-dependent readouts like column temperature, gas flows, voltages, etc per sample, which would be lost in the conversion. Fourthly, the raw data files are also used in TASQ for the annotation of the metabolites. While we appreciate the concerns of the reviewer, it is common practice in the metabolomics field to provide the original raw data files for open access repositories.

Although the authors clarified some aspects of the MS methodology in the response and the manuscript, but warning some are untracked changes: Like the following sentence: L539-541[Our metabolomics method is specifically tailored to analyze key polar metabolites of the most fundamental metabolic pathways with high confidence, instead of a broader untargeted method where identifications are more uncertain. I still found unclear the explanations and

description of the relationships between the actual MS data and analysis (that didn't have fragmentation spectra and ion mobility) and L545-546 "the Metabolite identification has been based on a combination of accurate mass, (relative) retention times, ion mobility data and fragmentation spectra, compared to the analysis of a library of standards". From the discussion, it seems that only the MS1 ion accurate mass and (relative?!) retention times were in fact used for the annotation of the present MS data, by matching them with those of internal standards from a library previously established (in other experimental conditions). Maybe what the authors are trying to indicate is that they previously analyzed the same biological matrix and identified the compounds using other experimental parameters (ion mobility and MS/MS) which indeed increase the confidence in the annotation. In any case, the MS acquisition and annotation method must be described more clearly in the manuscript.

Response: We believe that there is a misunderstanding in this matter. There are two steps to polar metabolomics: compound identification of the peaks and data acquisition. Both require different MS-settings and are performed separately on purpose. The first step, using standards, is not repeated for each new project as this is not only impractical, but also unnecessary. This is common practice in polar metabolomics, as many of the settings used for identification of some metabolites (e.g. ion mobility or MS/MS) interfere with the sensitivity needed for others (as opposed to for instance proteomics, where these steps are combined during acquisition). Additionally, due to the extreme diversity in the chemical composition of these metabolites, not all of these steps can be performed for each metabolite. For instance, it is a practical impossibility to fragment glycine or acquire a CCS value. Therefore, each identification is a pragmatic one, summarized in a value of confidence in the supplied literature reference. All data acquisition for metabolomics in this project has indeed been performed using the MS1 full scan of the Bruker Impact II, and included metabolites passed all manual quality control checks. Finally, the "relative retention time" in e.g. carnitine esters show a distinct pattern of retention times. It might be that their absolute retention time shifts, but their identification is very robust using the relative distances between each peak. The same goes for the different phosphorylation states of the nucleotides.

In light of this, we have amended the sentence to:

Methods, page 16. Line 559-566

Deleted:

~~*Metabolite identification has been based on a combination of accurate mass, (relative) retention times, ion mobility data and fragmentation spectra, compared to the analysis of a library of standards, which were performed in separate quality control experiments in the core facility.*~~

Added:

Additional confirmation of metabolite identity has been performed in separate experiments on a variety of instruments, based on a combination of accurate mass, (relative) retention times, ion mobility data and fragmentation spectra, compared to the analysis of a library of standards. MS1 results in this study were subsequently confirmed using the data of these prior analyses.

Reviewer #5 (Remarks to the Author):

In general, the prior comments have been adequately addressed. The statistical analyses are now clearly explained and more reproducible. However, although the authors have now added a identifier, it seems the study was registered after the actual start date. If this is indeed the case, I would suggest that the authors clearly state in the manuscript that the

study was retrospectively registered and include the dates of first study registration and first patient inclusion.

Response: The study was registered and accepted for online publication on clinicaltrials.gov before the first participant completed the study procedure. Since the initial start, we have added various amendments (including a one-year follow-up, additional participants and a new group of participants, i.e. ME/CFS), for which we amended the <https://classic.clinicaltrials.gov/ct2/history/NCT05225688> identifier. This can unintentionally be interpreted as a retrospective registration, which was not the case. We have added this to the sentence on line 368:

Methods, line 368

"The study protocol was approved by the medical ethics committee of the Amsterdam UMC (NL78394.018.21) and registered at www.clinicaltrials.gov (NCT05225688) before completion of the first participant."